# AutoGPS: Automated Geometry Problem Solving via Multimodal Formalization and Deductive Reasoning

**Bowen Ping, Minnan Luo[†], Zhuohang Dang, Chenxi Wang, Chengyou Jia**

Xi'an Jiaotong University

`315229706@stu.xjtu.edu.cn, minnluo@xjtu.edu.cn`

## Abstract

Geometry problem solving presents distinctive challenges in artificial intelligence, requiring exceptional multimodal comprehension and rigorous mathematical reasoning capabilities. Existing approaches typically fall into two categories: neural-based and symbolic-based methods, both of which exhibit limitations in reliability and interpretability. To address this challenge, we propose AutoGPS, a neuro-symbolic collaborative framework that solves geometry problems with concise, reliable, and human-interpretable reasoning processes. Specifically, AutoGPS employs a Multimodal Problem Formalizer (MPF) and a Deductive Symbolic Reasoner (DSR). The MPF utilizes neural cross-modal comprehension to translate geometry problems into structured formal language representations, with feedback from DSR collaboratively. The DSR takes the formalization as input and formulates geometry problem solving as a hypergraph expansion task, executing mathematically rigorous and reliable derivation to produce minimal and human-readable stepwise solutions. Extensive experimental evaluations demonstrate that AutoGPS achieves state-of-the-art performance on benchmark datasets. Furthermore, human stepwise-reasoning evaluation confirms AutoGPS's impressive reliability and interpretability, with 99% stepwise logical coherence.

## 1 Introduction

Mathematical reasoning constitutes a fundamental component of human intelligence. Among various mathematical domains, geometry problem solving (GPS) has attracted significant attention due to its intrinsic elegance and deductive completeness (Tarski, 1998; Trinh et al., 2024). Distinct from other mathematical disciplines such as algebra (Hendrycks et al., 2021; Cobbe et al., 2021; Yu et al., 2024), geometric problem solving inherently involves the analysis and comprehension of multimodal information, including visual diagrams and textual descriptions (Chen et al., 2022b; Seo et al., 2015; Lu et al., 2021; Wu et al., 2024; Peng et al., 2023; Huang et al., 2024). This process necessitates not only the understanding of multimodal problem formulation but also rigorous mathematical deduction to derive solutions (Lu et al., 2023; Gao et al., 2025; Huang et al., 2025b; Zhang et al., 2024a; Chen et al., 2022b; Liu et al., 2024a).

Recently, many methods have been proposed for this challenging task, which are primarily fall into two categories: (1) *Neural-based methods*, including specialized neural geometry solvers (Zhang et al., 2023; Li et al., 2024c) and multimodal large language models (MLLMs) (Gao et al., 2025; OpenAI, 2024; OpenAI et al., 2024; Qwen et al., 2025; Huang et al., 2025b; Chen et al., 2024c;b; Gao et al., 2024; Wang et al., 2024b; Chen et al., 2024a; Zhang et al., 2025; Sun et al., 2023; 2024), demonstrate superior multimodal comprehension capabilities. However, as shown in Figure 1 (left), such methods are **unreliable** and prone to generating plausible but logically flawed reasoning steps due to hallucination (Huang et al., 2025a), leading to erroneous conclusions. (2) *Symbolic approaches* (Lu et al., 2021; Peng et al., 2023; Wu et al., 2024) operate on formal language inputs through predefined rules and algebraic computations with basic mathematical rigor. However, as

---

[†]Corresponding Author.

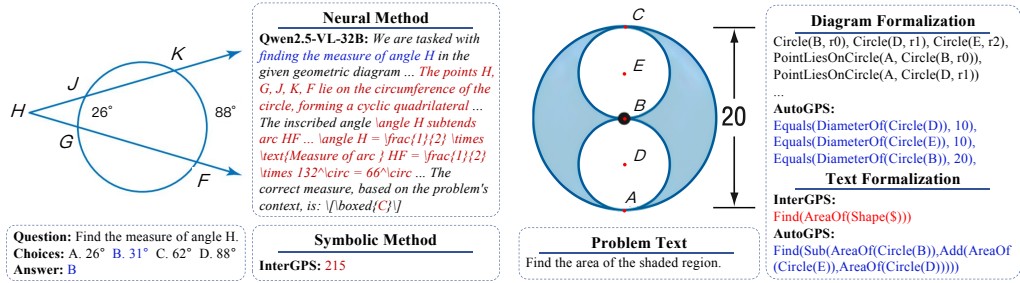

Figure 1: Failure cases of current methods. **Left:** Qwen2.5-VL-32B-Instruct exhibits hallucination-induced errors during reasoning, producing an incorrect conclusion. The symbolic method (Inter-GPS) also fails and lacks traceable steps for error diagnosis. **Right:** Inadequate cross-modal understanding leads to incomplete formalization, further obstructing symbolic solving. Blue/red annotations indicate correct/erroneous reasoning or answers.

exemplified in Figure 1 (right), they struggle to completely formalize given multimodal problem input, leading to **unreliable** problem-solving results. Furthermore, neural methods inherently **lack interpretability** (Zhang et al., 2021; Li et al., 2024c), and existing symbolic methods fail to provide **explicit human-interpretable** reasoning steps (Lu et al., 2021; Peng et al., 2023; Wu et al., 2024) as shown in Figure 1 (left). This naturally leads us to pose a significant research challenge: **Can we develop an automated geometry problem solving methodology that demonstrates reliable and interpretable reasoning capabilities?**

To address this challenge, we propose AutoGPS, a neuro-symbolic collaborative framework that solves geometry problems with concise, reliable, and human-interpretable reasoning processes. Specifically, AutoGPS comprises two principal components: the Multimodal Problem Formalizer (MPF) and the Deductive Symbolic Reasoner (DSR). The MPF is a formalization agent that processes image-text pairs of geometric problems. By enhancing multimodal large language models with domain-specific parsing tools, it executes pre-formalization and multimodal alignment operations to transform geometric problems into formal language representations. The DSR is a symbolic computation engine based on predefined rules, accepting formal problem descriptions as input. It validates the formalization and provides feedback to the MPF for refinement until the formalization is error-free. Subsequently, the DSR formulates the problem-solving process as a hypergraph expansion task, where the formal language literals are

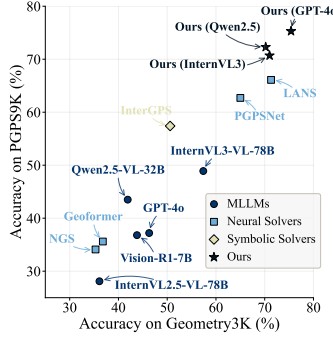

Figure 2: Performance comparison among existing methods.

treated as nodes and derivation steps are treated as hyperedges. It iteratively applies two complementary strategies to expand the hypergraph until the solution node is reached. Upon solution attainment, it identifies the minimal sub-hypergraph for solution derivation and generates a syllogistic-structured reasoning process. Therefore, the proposed AutoGPS effectively solves the geometry problem with rigorous, concise, and human-readable derivation, maintaining both reliability and interpretability.

Experimental results on two GPS benchmarks (Lu et al., 2021; Zhang et al., 2023) demonstrate that our AutoGPS outperforms the existing methods by margins of 4.1% and 9.2% and the state-of-the-art mathematical reasoning MLLMs by 18.0% and 26.4%. Human stepwise reasoning evaluations validate that AutoGPS achieves 99% stepwise accuracy, surpassing the 71% accuracy of the best-performing MLLM (Zhu et al., 2025), demonstrating absolute superiority in reliability and interpretability over existing methods.

## 2 RELATED WORK

### 2.1 MATHEMATICAL REASONING WITH MLLMS

Recent advancements in MLLMs have demonstrated significant potential for mathematical reasoning in visual contexts, as evidenced by multiple benchmark studies. The research community has

developed specialized datasets to systematically assess these capabilities: MathVista (Lu et al., 2023) serves as the pioneering benchmark for evaluating visual mathematical reasoning, while MATH-Vision (Wang et al., 2024a) provides comprehensive assessments across core mathematical domains, including algebra, statistics, and geometry. MathVerse (Zhang et al., 2024b) contributes 2,612 human-annotated visual mathematical problems spanning planar geometry, solid geometry, and functional analysis. Complementing these efforts, GeoEval (Zhang et al., 2024a) specifically targets geometric reasoning evaluation through plane and solid geometry problems. Building upon these foundational works, our research involves a comprehensive evaluation of MLLMs' performance in plane geometry contexts. We utilize two larger-scale datasets: Geometry3K (Lu et al., 2021) and PGPS9K (Zhang et al., 2023), with particular emphasis on both formalization and solution-generation.

## 2.2 GEOMETRY PROBLEM SOLVING

Geometry problem solving remains a critical research focus (Lu et al., 2021; Peng et al., 2023; Wu et al., 2024; Li et al., 2024c; Gao et al., 2025; Huang et al., 2025b; Seo et al., 2015; Hao et al., 2022; Zhang et al., 2022; 2023; Sun et al., 2025), with existing approaches falling into three main categories: neural-based, symbolic-based, and neural-symbolic methods. Neural methods leverage trained networks (PGPSNet (Zhang et al., 2023) and LANS (Li et al., 2024c)) or fine-tuned multimodal LLMs (LLaVA (Liu et al., 2023; 2024a;b), G-LLaVA (Gao et al., 2025), Vision-R1 (Huang et al., 2025b)) for geometric reasoning, while symbolic systems (Inter-GPS (Lu et al., 2021), E-GPS (Wu et al., 2024)) employ formal language (Xu et al., 2024b;a) and deductive rules. Neural-symbolic approaches such as GeoDRL (Peng et al., 2023) and FGeo-HyperGNet (Zhang et al., 2024c) integrate neural networks with symbolic solvers for heuristic search, reducing the search space to enhance efficiency. These methods, however, exhibit limitations in reliability and interpretability. Neural methods suffer from hallucination issues, leading to incorrect reasoning and answers, while symbolic methods produce algebraic solutions lacking procedural transparency. Our framework first translates multimodal problems into formal language representations, then executes symbolic deduction to generate human-readable reasoning steps, ensuring reliability and interpretability.

## 2.3 NEURAL-SYMBOLIC METHODS FOR MATHEMATICS

The limitations of purely neural (unreliable) and purely symbolic (non-scalable) methods have driven the development of neuro-symbolic frameworks for mathematical reasoning (Trinh et al., 2024; Li et al., 2025; 2024b; Wu et al., 2025; Shang et al., 2025; Singh et al., 2025; Wu & Yu, 2025). Specifically, these approaches are utilized for complex problem-solving by leveraging neural methods for heuristic guidance in tasks like geometry proof and inequality deduction (Trinh et al., 2024; Li et al., 2025). They are also employed for reliable dataset generation, ensuring the logical rigor of mathematical problem-proof pairs via symbolic verification (Li et al., 2024b; Wu et al., 2025). Another key direction involves agent-style methods, which empower LLMs with planning and self-correction to optimally interact with external symbolic tools for enhanced precision and robustness (Yang et al., 2023; Shang et al., 2025; Singh et al., 2025; Wu & Yu, 2025). These methods influence our approach. The proposed AutoGPS utilizes an agent-style mechanism in its MPF coupled with a feedback and refinement loop involving the DSR. This collaboration ensures precise formalization and guarantees that the DSR's output is highly reliable and interpretable, providing minimal, syllogistic-structured human-readable reasoning steps.

# 3 METHODOLOGY

## 3.1 PROBLEM FORMULATION

Following (Lu et al., 2021), given a predefined theorem set $\mathcal{T}$, we aim to solve geometry problems formalized as $\mathcal{D} = \{(D_i, T_i)_{i=1}^N\}$. Here, $(D_i, T_i)$ is the $i$-th multimodal geometry problem description, where $D_i$ represents the original geometric diagram and $T_i$ denotes the corresponding textual description, specifying the problem objective (e.g., "Find length of line AB"). In the following, we omit the subscript $i$ to represent $(D_i, T_i)$ as $(D, T)$ for brevity. To ensure the reliability and interpretability, given $(D, T)$, our AutoGPS aims to generate the correct solution $a^*$ with corresponding stepwise reasoning process $S = \{s_1, s_2, \ldots, s_n\}$. where $s_i$ is the $i$-th reasoning step.

## 3.2 MODEL OVERVIEW

The proposed AutoGPS framework, as illustrated in Figure 3, comprises two core components: the Multimodal Problem Formalizer (MPF) and the Deductive Symbolic Reasoner (DSR). Initially, given formal language space $\mathcal{L}$ defined in Appendix F, the MPF formalizes input geometric problems $(D, T)$ into structured formal language representations $L = \{l_1, l_2, \ldots, l_k, \text{Find}(t)\} \subset \mathcal{L}$. Here each $l_i$ is a formal language literal (e.g., $\text{Line}(A, B)$), $\text{Find}(t)$ is a statement indicating the problem-solving goal $t$ (e.g., $\text{LengthOf}(\text{Line}(A, B))$). Given formalization $L$, the DSR validates the self-consistency of $L$ and provides feedback to the MPF for refinement iteratively until the formalization is error-free. Then, the DSR formulates the problem-solving process as a hypergraph expansion task, where the formal language literals are treated as nodes and derivation steps are treated as hyperedges. Subsequently, DSR iteratively expands the hypergraph through two complementary strategies, *Deductive Reasoning* and *Algebraic Reasoning*, until the solution literal $a^*$ is reached. Here $a^*$ is of form $\text{Equals}(t, v)$, where $t$ is the target literal and $v$ is the solution value (e.g., $\text{Equals}(\text{LengthOf}(\text{Line}(A, B)), 2)$). Finally, DSR generates a human-readable and concise solution consisting of a sequence of reasoning steps $S = \{s_1, s_2, \ldots, s_n\}$ based on the minimal reasoning sub-hypergraph leading to $a^*$. The details of each component are described in the following sections.

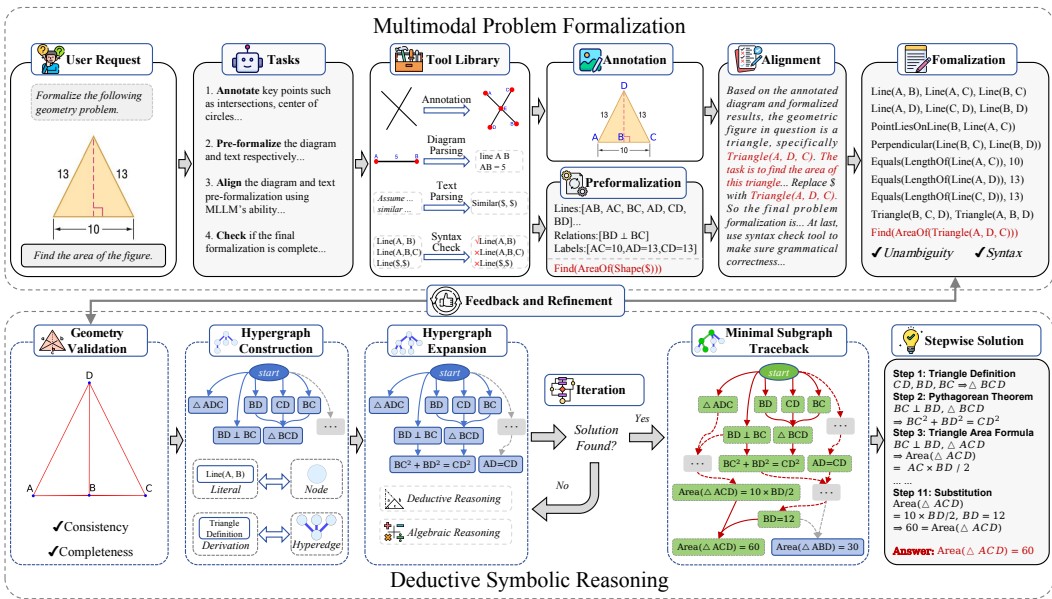

Figure 3: Overview of the proposed AutoGPS framework.

## 3.3 MULTIMODAL PROBLEM FORMALIZER

To transform geometry problems into formal language representations, we propose the Multimodal Problem Formalizer (MPF). Specifically, the MPF is a multimodal agent equipped with a tool library and executes a predefined sequence of tasks (*Annotation*, *Pre-formalization*, and *Multimodal Alignment*) to generate a complete formal language representation $L$.

**Annotation.** Formal language specifications depend on point labels, while real-world geometric diagrams often lack annotations of critical points. To address this issue, we introduce a pre-trained model $M_a$ based on the FPN architecture (Lin et al., 2017) to detect key points and explicitly annotate them in the diagram. For a given geometry diagram $D$, the model $M_a$ detects key points such as intersection points and circle centers, and assigns labels to these points, formulated as $P_t, Y = M_a(D)$, where $P_t$ represents the detected key points (e.g., triangle vertices in Figure 3) and $Y$ refers to the corresponding label annotations (e.g., $ABCD$ in Figure 3). Then, a new diagram $D' = (D, P_t, Y)$ is constructed with all key points annotated for subsequent processing.

**Pre-formalization.** Given the annotated diagram $D'$, we introduce a pre-formalization process to extract geometric relations. However, Figure 1 (left) shows that existing MLLMs still struggle with

extracting precise geometric relations. To address these, we introduce a *Diagram Parser* and a *Text Parser* to parse the information in the diagram $D'$ and the problem text $T$ respectively.

To precisely obtain the information in the diagram, we introduced a pre-trained neural model $M_d$ based on PGDP-Net (Zhang et al., 2022) as the *Diagram Parser*. For a given annotated diagram $D'$, the model $M_d$ extracts geometric primitives (e.g., segments, circles) and relations (e.g., parallelism, perpendicularity), formulated as $P_g, R = M_d(D')$, where $P_g$ represents geometric primitives (e.g., lines shown in Figure 3) and $R$ represents geometric relations (e.g., BD $\perp$ BC shown in Figure 3).

Inspired by Lu et al. (2021), for better efficiency, we utilize a rule-based *Text Parser* $M_t$ to translate the natural language text $T$ into formal language literals $T' = \{l'_1, l'_2, \ldots, l'_{k'}, \text{Find}(t')\}$, where each $l'_i$ is a pseudo formal language literal and $\text{Find}(t')$ is a pseudo problem solving goal. Here, "pseudo" indicates that the literals are not yet in the formal language space $\mathcal{L}$, which may include incomplete expressions (e.g., $\text{Shape}(\$)$ in Figure 3 with the unknown represented as $). Unlike neural sequence-to-sequence approaches (Vaswani et al., 2017; Gan & Yu, 2018), rule-based methods empirically show superior efficiency and better performance without relying on large-scale datasets and computational resources (Seo et al., 2015; Bansal et al., 2014; Lu et al., 2021). However, previous rule-based methods (Seo et al., 2015; Lu et al., 2021) struggle with handling complex mathematical expressions (e.g., chain of equalities like $x_1 = x_2 = \ldots = x_n$). To address this, we extend the pattern-matching rules of the *Text Parser* to transform complex expressions into formal language literals. Finally, the pre-formalization $F$ is formulated as $F = (P_g, R, T')$ for further processing.

**Multimodal Alignment.** Since the global geometric information may not be fully captured in the pre-formalization stage (e.g., shaded region and diameters in Figure 1 (right)), we further introduce the multimodal alignment phase to enhance the completeness of the formalization. Given the annotated diagram $D'$ and the pre-formalization $F$, we leverage MLLMs' multimodal understanding capabilities to generate a complete formalization set $L$ by aligning $F$ with $D'$. Specifically, the multimodal agent: (1) takes $D'$ and $F$ as input, then (2) analyzes multimodal information to clarify ambiguities and fill in missing information, and (3) generates a complete formalization set $L = \{l_1, l_2, \ldots, l_k, \text{Find}(t)\}$. Finally, the MPF outputs the complete formalization set $L$ to the DSR for further processing.

Compared to existing methods which only parse elementary geometric relationships, the proposed MPF effectively comprehensively extracts fine-grained and global geometric relationships from multimodal information, thereby enhancing the completeness and accuracy of the problem formalization.

## 3.4 DEDUCTIVE SYMBOLIC REASONER

Given the geometry problem formalization $L = \{l_1, l_2, \ldots, l_k, \text{Find}(t)\}$, to derive the solution $a^*$ with a reasoning process $S = \{s_1, s_2, \ldots, s_n\}$, we propose the Deductive Symbolic Reasoner (DSR). Specifically, the DSR is a symbolic engine that formulates the problem-solving process as a hypergraph expansion task, and generates human-readable reasoning steps $S$ leading to $a^*$.

**Geometry Validation.** Due to the input sensitivity of rule-based symbolic solvers (Lu et al., 2021), we propose a geometry validation step to ensure the consistency and completeness of the formalization $L$. Specifically, the DSR constructs a symbolic representation $d$ of the formalization $L$, and verifies whether these relations are self-consistent and complete. If the formal descriptions are incomplete but can be logically completed using existing knowledge, the symbolic engine automatically supplements the missing relations. For instance, given PH $\perp$ AB with point H on AB, the DSR infers other two missing relations, *i.e.*, PH $\perp$ AH, and PH $\perp$ BH. Moreover, contradictory expressions, such as asserting collinearity of points A, B, C while simultaneously including $\text{Triangle}(A, B, C) \in L$, will be flagged as inconsistent. Consistent and complete formalization should provide equivalent information given in the original diagram $D$. Inconsistent formalizations with error messages will be sent back to the MPF for refinement and resubmission iteratively, until the DSR receives a self-consistent formalization. If the maximum iteration threshold is reached, the DSR will terminate the process, and the problem will be marked as unsolvable. Finally, a complete and consistent symbolic representation $d = \{l^d_1, l^d_2, \ldots, l^d_n\}$ is constructed for the next step, hypergraph reasoning.

**Hypergraph Construction.** Prior symbolic solvers (Lu et al., 2021; Peng et al., 2023; Wu et al., 2024) derive solutions by constructing algebraic equation systems between known and unknown quantities, expanding these systems via theorems, and solving them algebraically. However, this approach exhibits limited interpretability due to the lack of traceability in the algebraic solving process.

Inspired by AlphaGeometry (Trinh et al., 2024), we propose a hypergraph-based deductive reasoning framework that preserves full reasoning traceability. Specifically, we enforce each reasoning step $s_i \in S$ following a syllogistic structure: (1) a theorem $t \in \mathcal{T}$, (2) a premise set $P = \{p_1, p_2, \ldots, p_m\}$ including $m$ premise literals $p_i$, and (3) a conclusion set $Q = \{q_1, q_2, \ldots, q_{m'}\}$ including $m'$ conclusion literals $q_i$. Each reasoning step is formulated as $P \xrightarrow{t} Q$. For example, applying the Pythagorean theorem can be represented as: $\{\triangle ABC, AB \perp AC\} \xrightarrow{\text{Pythagorean Theorem}} \{AB^2 + AC^2 = BC^2\}$. Then, we model each literal as a node and each derivation step as a hyperedge. The reasoning process is represented as a hypergraph $G = (V, E)$, where $V$ is the node set (all known literals) and $E$ is the hyperedge set (all derivation steps). Given symbolic representation $d$ and problem formalization $L$, the initial hypergraph is constructed with a hyperedge labeled "Known Facts" connecting the *start* node (a trivial root) to all literals $l_i \in L$ and $l_i^d \in d$.

**Hypergraph Expansion.** While AlphaGeometry (Trinh et al., 2024) excels at proving geometric theorems, its core framework is strictly confined to deductive reasoning (DD) and geometric relation transitions (AR), leaving general algebraic equation solving outside of its scope. To address this limitation, we propose a unified hypergraph expansion framework that integrates both deductive and algebraic reasoning. To derive the solution literal $a^*$ from the initial hypergraph $G$, we expand $G$ through the following two complementary strategies: (1) *Deductive Reasoning* matches each theorem $t \in \mathcal{T}$ across node set $V$ to expand $G$ with new hyperedges (reasoning steps) and conclusion nodes. (2) *Algebraic Reasoning* solves algebraic equations through stepwise transformations instead of bulk solving (Lu et al., 2021; Peng et al., 2023; Wu et al., 2024), to ensure that each computation step is intuitively human-interpretable. This strategy implements four atomic operations: *Equivalent Substitution*, *Constant Evaluation*, *Univariate Non-Linear Equation Solving*, and *Linear Equation System Solving*. Each algebraic transformation is formulated as a special deductive operation. For example: Equivalent substitution deriving $a = \sin(x)$ from $a = b$ and $b = \sin(x)$ is represented as:

$$\{a = b, b = \sin(x)\} \xrightarrow{\text{Equivalent Substitution}} \{a = \sin(x)\} \tag{1}$$

This uniform hyperedge representation ensures coherent graph expansion. To eliminate redundant premise nodes in hyperedges, we enforce minimal sufficient equation sets for each atomic operation. This is straightforward for the first three operations. For *Linear Equation System Solving*, which yields multiple new equations, we solve a mixed-integer linear programming problem to identify the minimal sufficient equation set required to derive each new equation.

The DSR iteratively applies *Deductive Reasoning* and *Algebraic Reasoning* alternately to expand the hypergraph until either the solution node $a^*$ is identified or a predefined maximum iteration threshold is reached. Additionally, to prevent cyclic reasoning during the process, the DSR tracks all predecessors for each node (including all its premise nodes and their recursive predecessors) to block backward reasoning toward ancestral nodes. This mechanism ensures the entire reasoning hypergraph remains a *directed acyclic hypergraph*. This unified hypergraph-based framework guarantees consistent formulation of diverse reasoning processes while maintaining full traceability and interpretability throughout the problem-solving trajectory.

**Minimal Solution Generation.** Once identifying the solution node $a^*$, the DSR generates the most concise solution steps by extracting the minimal reasoning sub-hypergraph that connects the *start* node to the $a^*$ node. This subgraph must satisfy the following criteria:

(1) *Single Source*: Contains exactly one source node (in-degree of zero), *i.e.*, the initial node *start*;

(2) *Single Sink*: Contains exactly one sink node (out-degree of zero), *i.e.*, the solution node $a^*$;

(3) *Minimality*: Among all subgraphs satisfying (1) and (2), it minimizes the number of hyperedges.

This optimization problem, formerly known as the *hypergraph shortest path problem*, is generally NP-hard but solvable in polynomial time for directed acyclic hypergraphs like ours (Gallo et al., 1993; Gao et al., 2014). Once the minimal sub-hypergraph $G_{min}$ is obtained, the DSR first performs topological sorting on $G_{min}$ to establish dependency order, then translates the sorted hyperedge sequence into a human-readable stepwise solution $S = \{s_1, s_2, \ldots, s_n\}$. This process ensures logical coherence and preserves interpretability throughout the solution trajectory. Appendix E provides some examples of minimal reasoning hypergraphs. The complete algorithm is detailed in Algorithm 1.

---

**Algorithm 1** Deductive Symbolic Reasoning Algorithm

---

**Input:** problem formalization $L = \{l_1, l_2, \ldots, l_k, \text{Find}(t)\}$, max iteration $n$
**Output:** solution literal $a^*$, minimal stepwise solution $S$
 1: $d \leftarrow \text{GeometryValidation}(L)$          ▷ If inconsistent, provide feedback to MPF and return
 2: $G \leftarrow \text{ProofGraph}(L, d)$          ▷ Hypergraph Construction
 3: $a^* \leftarrow null, i \leftarrow 0$
 4: **while** $a^* = null$ and $i < n$ **do**
 5:     $G \leftarrow \text{DeductiveReasoning}(G, \mathcal{T})$          ▷ Deductive Reasoning
 6:     $G \leftarrow \text{AlgebraicReasoning}(G)$          ▷ Algebraic Reasoning
 7:     $a^* \leftarrow \text{MatchLiteral}(\text{Equals}(t, v))$          ▷ Check if solution is found
 8:     $i \leftarrow i + 1$
 9: **end while**
10: $G_{\min} \leftarrow \text{FindMimimalReasoningSubgraph}(G, a^*)$
11: $S \leftarrow \text{TopologicalSort}(G_{\min})$          ▷ Minimal Solution Generation
12: **return** $a^*, S$

---

## 4 EXPERIMENTS

### 4.1 EXPERIMENTAL SETUP

**Datasets and Evaluation.** Experiments were conducted on two specialized GPS benchmarks: Geometry3K (Lu et al., 2021) and PGPS9K (Zhang et al., 2023), which contain 3,001 and 9,000 image-text pairs of plane geometric problems, respectively, both providing four candidate options and corresponding ground-truth answers. To enable comprehensive performance comparison, we adopted both *Choice* and *Completion* evaluation formats. Additionally, we utilized *Stepwise Accuracy* evaluated by human experts for stepwise reasoning quality. More details are provided in Appendix A.

**Implementation Details.** We employed several state-of-the-art MLLMs as agents in our framework. Open-source: Qwen2.5-VL-32B-Instruct (Qwen et al., 2025) (Qwen2.5), InternVL3-78B (Zhu et al., 2025) (InternVL3). Closed-source: GPT-4o (OpenAI, 2024). See Appendix A.3 for more details.

Table 1: Performance comparison among state-of-the-art geometry problem solvers.

| Method | Geometry3K | | PGPS9K | |
|---|---|---|---|---|
| | *Choice* | *Completion* | *Choice* | *Completion* |
| *MLLMs* | | | | |
| G-LLaVA-13B | 29.0 | 0.3 | 27.0 | 0.0 |
| Vision-R1-7B | 57.1 | 43.8 | 49.6 | 36.8 |
| Qwen2.5-VL-32B | 67.6 | 41.9 | 56.1 | 43.5 |
| InternVL2.5-78B | 60.9 | 36.1 | 51.3 | 28.1 |
| InternVL3-78B | 74.5 | 57.4 | 61.1 | 48.9 |
| GPT-4o | 57.1 | 46.3 | 46.0 | 37.2 |
| *Neural Solvers* | | | | |
| NGS | 58.8 | 35.3 | 46.1 | 34.1 |
| Geoformer | 59.3 | 36.8 | 47.3 | 35.6 |
| PGPSNet | 77.9 | 65.0 | 70.4 | 62.7 |
| LANS | **82.3** | 71.3 | 73.8 | 66.1 |
| *Symbolic Solvers* | | | | |
| InterGPS | 63.5 | 50.6 | 66.2 | 57.4 |
| E-GPS | 67.9 | - | - | - |
| GeoDRL | 68.4 | - | - | - |
| Ours (InternVL3) | 77.6 | 70.2 | 79.2 | 72.3 |
| Ours (Qwen2.5) | 78.2 | 71.0 | 78.0 | 70.7 |
| Ours (GPT-4o) | 81.6 | **75.4** | **81.5** | **75.3** |

**Baselines.** (1) MLLMs: G-LLaVA-13B (Gao et al., 2025), Vision-R1-7B (Huang et al., 2025b), Qwen2.5-VL-32B-Instruct, InternVL2.5-78B (Chen et al., 2024c), InternVL3-78B and GPT-4o. (2) Specialized neural solvers: NGS (Chen et al., 2022b), Geoformer (Chen et al., 2022a), PGPSNet (Zhang et al., 2023) and LANS (Li et al., 2024c). (3) Symbolic solvers: InterGPS (Lu et al., 2021), GeoDRL (Peng et al., 2023) and E-GPS (Wu et al., 2024).

### 4.2 EXPERIMENTAL RESULTS

This section presents the main experimental results of the proposed AutoGPS framework. We provide additional experiments and analyses in Appendix B, including: (1) answer reliability evaluation, (2) symbolic solver comparison, (3) failed cases analysis and (4) efficiency analysis.

**Performance Comparison with State-of-the-Art Methods.** As shown in Table 1, our AutoGPS achieves competitive performance across task formats and datasets. For *Choice* tasks, it attains comparable accuracy to SOTA methods on Geometry3K (81.6% vs. 82.3%), while demonstrating a 7.7% improvement on the more complex PGPS9K dataset. In *Completion* tasks, our framework outperforms SOTA methods by 4.1% and 9.2% on Geometry3K and PGPS9K, respectively.

Notably, most of the neural methods exhibit significant performance degradation in open-form *Completion* tasks (e.g., Qwen2.5-VL-32B dropping by 25.7% on Geometry3K and 12.6% on PGPS9K), as their probabilistic guessing mechanisms struggle to constrain solution spaces without predefined options. In contrast, our AutoGPS demonstrates superior stability across task formats and dataset complexities, maintaining minimal accuracy gaps between *Choice* and *Completion* tasks ($\Delta$=6.2% on both datasets), attributed to its rigorous symbolic reasoning that derives solutions through explicitly defined rules rather than data-driven predications.

**Performance Comparision with Specialized MLLM.** G-LLaVA-13B and Vision-R1-7B are specialized MLLMs fine-tuned for geometry problem solving. While Vision-R1-7B achieves comparable accuracy with a small-scale model, G-LLaVA-13B struggles across all tasks (27.0–29.0% on *Choice*, near-zero on *Completion*), revealing a critical limitation: *such methods that rely on exhaustive textual descriptions fail when text-diagram alignment is not perfect, which is common in real-world problems.* In contrast, our AutoGPS framework fundamentally addresses this gap by effectively extracting geometric relations from diagrams and leveraging texts as supplementary information. This methodology enables a thorough understanding of geometry problems through multimodal integration, thereby attaining enhanced robustness in geometric comprehension.

**Reasoning Reliability Evaluation.** Generative large models suffer from hallucination phenomena, leading to cascading errors in reasoning chains. In contrast, AutoGPS employs symbolic syllogistic reasoning to enforce rigorous derivations, inherently avoiding such issues. To quantify this advantage, we conducted a human evaluation experiment and utilized the *Stepwise Accuracy* metric to assess the reasoning reliability of different methods, as shown in Figure 4. From the subset of problems consistently and correctly solved by all methods, we randomly selected 100 instances to exclusively focus on the reliability of their reasoning processes. Despite equivalent final accuracy, MLLMs exhibited substantial logical flaws (Qwen2.5: 33.0%, InternVL3: 29.0%, GPT-4o: 32.0%), whereas AutoGPS achieved almost perfect logical coherence (99%) across all cases. This demonstrates AutoGPS's capability in simultaneously guaranteeing answer correctness and reasoning validity, thereby establishing both interpretability and reliability. Figure 6 illustrates a reasoning process comparison example. Additional comparative examples are provided in Appendix D.2.

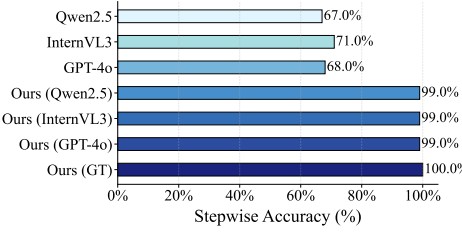

Figure 4: Reasoning reliability evaluation.

Figure 5: Formalization quality comparison.

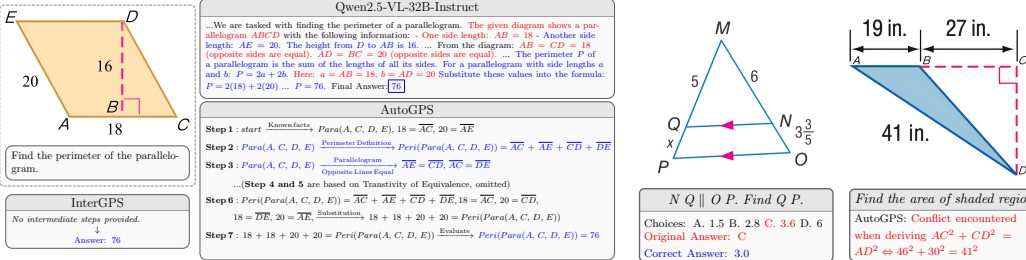

Figure 6: Reasoning process Comparision. Blue/Red indicates correct/wrong reasoning steps.

Figure 7: AutoGPS robustly handles noisy inputs.

**Formally Grounded Reasoning Enhances Noise Robustness.** The inherent rigor of syllogistic reasoning endows AutoGPS with exceptional resilience against inconsistent problem statements. As demonstrated in Figure 7 (left), the framework successfully solves problems with missing valid options through autonomous derivation of ground truth solutions. The complete reasoning process is detailed in Appendix D.1, with corresponding hypergraph representations visualized in Figure 20. In Figure 7 (right), it detects geometric contradictions (e.g., a right triangle leg exceeding hypotenuse

length: $19 + 27 = 46 > 41$) during the reasoning process. These results validate the robustness of DSR against noisy inputs, demonstrating its appealing potential for real-world applications such as in education and tutoring systems, where robustness and reliability are important.

**Formalization Quality Comparison.** Symbolic solvers are highly sensitive to the formalization quality. We evaluate our AutoGPS against InterGPS (Lu et al., 2021) on Geometry3K through two metrics: (1) *Jaccard Similarity* measures overlap between generated and ground-truth formalizations, (2) *Completion Accuracy* assesses DSR's problem-solving success using formalizations as input. As demonstrated in Figure 5, AutoGPS achieves a Jaccard similarity of 0.868, significantly higher than InterGPS's 0.395, indicating substantially more accurate formal representations. This improvement translates to a 27.6% increase in completion accuracy, showcasing that AutoGPS's formalization captures more complete and accurate geometric relations by *Pre-formalization*'s guidance and *Multimodal Alignment*'s refinement, thereby enhancing the overall problem-solving performance.

## 4.3 ABLATION STUDY

To systematically evaluate component efficacy within our framework, we performed ablation studies targeting both the MPF and DSR. Specifically, we analyzed the impact of *Pre-formalization* and *Multimodal Alignment* in the MPF module, and the effectiveness of *Minimal Solution Generation* in the DSR module. Additional ablation studies on the DSR module are provided in Appendix B.

**Pre-formalization.** We compared the performance of AutoGPS with and without *Pre-formalization* on the *Completion* task. When pre-formalization is not applied, the model generates formalizations directly from annotated diagrams and problem texts. The results are shown in Figure 8, where the *Completion* accuracy is significantly improved by 50.9%–62.4% across all models. Since MLLMs still struggle to identify precise geometric relationships such as collinearity shown in Figure 1 (left), the generated formalizations are often inaccurate. The pre-formalization guides them to capture accurate geometric configurations, providing formalizations of higher quality.

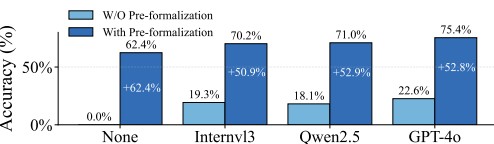

Figure 8: *Completion* performance with/without *pre-formalization*.

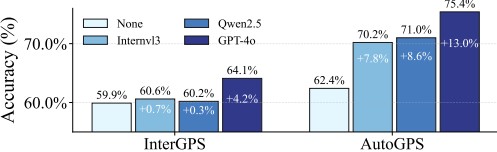

Figure 9: *Completion* performance with different alignment models.

**Multimodal Alignment.** We evaluated problem-solving accuracy across alignment configurations, comparing no alignment with three distinct multimodal alignment methods (Figure 9). Experiments show that applying multimodal alignment to pre-formalization achieved performance improvements ranging from 7.8% to 13.0%, when evaluated with our symbolic solver. Notably, even for solvers such as InterGPS that demonstrated primary capability in handling ambiguous formalizations, multimodal alignment maintained a 4.2% improvement in problem-solving accuracy. This demonstrates the effectiveness of our multimodal alignment phase to address semantic ambiguities and capture global geometric relationships, thereby providing more complete formalizations.

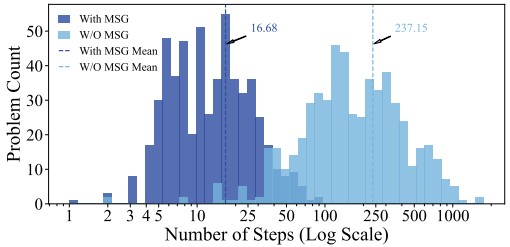

Figure 10: Steps distribution analysis for solution generation with versus without minimal solution generation (MSG).

**Minimal Solution Generation.** With Geometry3K ground-truth formalization, we analyzed the reasoning step distribution with and without minimal solution generation, as visualized in Figure 10. The framework achieves a substantial step reduction, decreasing the average number of reasoning steps from 237.15 to 16.69 (93% reduction). It demonstrates that minimal solution generation systematically identifies the optimal deduction trajectory, resulting in a more concise solution without sacrificing mathematical rigor.

## 5 CONCLUSION

This study introduces AutoGPS, a neural-symbolic framework that addresses key challenges in automated geometry problem-solving through multimodal formalization and deductive reasoning. By aligning diagram-text semantics and generating verifiable stepwise proofs, AutoGPS outperforms existing approaches on benchmark datasets while ensuring reliability and interpretability.

## ACKNOWLEDGEMENTS

This work is supported by the Fundamental and Interdisciplinary Disciplines Breakthrough Plan of the Ministry of Education of China (No. JYB2025XDXM101), the National Natural Science Foundation of China (No. 62272374, No. 62192781), the Natural Science Foundation of Shaanxi Province (No.2024JC-JCQN-62), the State Key Laboratory of Communication Content Cognition under Grant No. A202502, the Key Research and Development Project in Shaanxi Province (No. 2023GXLH-024).

## ETHICS STATEMENT

This research aims to advance multimodal geometry problem solving for academic and educational purposes. All experiments are conducted on publicly available datasets without sensitive information. We see minimal risk of misuse, but recommend responsible use in practice.

## REPRODUCIBILITY STATEMENT

We provide all code and data necessary to reproduce our main experiments in the supplementary materials. The detailed experimental setup is described in Section 4.1 and Appendix A. Our codebase will be made publicly available to facilitate further research.

## LLM USAGE STATEMENT

LLMs were used for (1) language polishing and (2) as multimodal agents inside our AutoGPS pipeline. Specific models, versions, hyper-parameters are provided in the Section 4.1 and Appendix A.3. The prompt templates are provided in Appendix G. All LLM outputs incorporated into the manuscript were reviewed and verified by the authors; authors accept full responsibility for the final content. LLMs were not listed as authors.

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

## A  MORE EXPERIMENTAL SETUP DETAILS

### A.1  DATASET

We evaluated our AutoGPS on two datasets: Geometry3K (Lu et al., 2021) and PGPS9K (Zhang et al., 2023). Geometry3K contains 3,001 geometry problems with ground-truth formalization labeled and double-checked by our human experts. The ground-truth formalization was used for two purposes: (1) to evaluate the quality of generated formalization results, (2) as the input of symbolic solvers to evaluate the performance of symbolic solvers.

### A.2  EVALUATION METRICS

**Choice and Completion Accuracy.** To enable comprehensive performance comparison, we adopted both *Choice* and *Completion* evaluation formats. For *Choice* tasks, the option answered by the model or numerically closest to the solver's output is selected, with random selection from four options if resolution fails. For *Completion* tasks, answers are strictly validated through numerical equivalence to ground-truth values, with unresolved cases automatically classified as incorrect. For all tasks involving MLLMs, performance was evaluated using the Pass@3 metric, which deems a task successful if at least one out of three independent sampling attempts produces a correct solution.

**Formalization Quality.** To evaluate the quality of formalization, we employed Jaccard similarity (Jaccard, 1901) as one of the metrics. It is defined as the size of the intersection divided by the size of the union of two sets, specifically:

$$J(P, Y) = \frac{|P \cap Y|}{|P \cup Y|},$$

where $P$ is the set of predicted formalization results and $Y$ is the set of ground-truth formalization results. The order of literals in the formalization does not matter.

**Human Evaluation of Stepwise Reasoning.** To evaluate the quality of stepwise reasoning, we employed *Stepwise Accuracy*. When evaluated by *Stepwise Accuracy*, a reasoning process is correct if and only if all reasoning steps are logically valid and lead to a correct answer. Our evaluation is conducted on the subset of problems that are consistently and correctly solved by all methods, to isolate the impact of final answer correctness. These problems with corresponding reasoning processes were distributed to three human experts for evaluation. The detailed criteria of stepwise reasoning evaluation are provided as follows:

1. **Accuracy of Geometric Information Comprehension**: Whether the textual and graphical information in the problem has been properly captured, including geometric elements (e.g., points, lines, circles), geometric relationships (e.g., perpendicularity, intersections), and text labels (whether they are correctly associated with the corresponding geometric elements).

2. **Correctness of Theorem Application**: Whether each theorem is applied to appropriate geometric entities and derives valid conclusions.

3. **Validity of Algebraic Transformations**: Whether algebraic operations are correctly implemented and yield accurate equations.

4. **Logical Coherence and Consistency**: Whether the reasoning exhibits logical coherence and consistency. Intermediate conclusions must align with the final answer, with no critical steps omitted.

All reasoning steps were independently examined by three human experts. For contested items (i.e., those with divergent evaluations), the correctness of the reasoning process was ultimately determined through collective deliberation and majority voting.

## A.3 IMPLEMENTATION DETAILS

The symbolic method InterGPS was reproduced using the authors' open-source code. while results for GeoDRL (Peng et al., 2023) and E-GPS (Wu et al., 2024) were extracted from original publications due to code unavailability. All experiments were executed on an Intel(R) Xeon(R) Gold 6226R CPU @ 2.90GHz platform in conjunction with eight NVIDIA GeForce RTX 3090 GPUs. A strict timeout threshold of 1800 seconds was imposed on symbolic solvers, where any computation exceeding this duration was systematically categorized as resolution failure. For MLLMs hyperparameters, we set the max_tokens to 3096, temperature to 0.1, and top-p to 1.

## B ADDITIONAL EXPERIMENTAL RESULTS

**Answer Reliability Analysis.** To further evaluate the reliability of answers generated by different methods, we computed the *Answer Reliability Rate* (ARR) on the *Completion* task, defined as follows:

$$\text{ARR} = \frac{N_{\text{correct}}}{N_{\text{valid}}}, \tag{2}$$

where $N_{correct}$ denotes the number of correct answers and $N_{valid}$ refers to the number of valid answers produced by the model. This metric reflects the trustworthiness of a given answer, *i.e.*, the probability that a returned answer is correct. For MLLMs, a valid answer is defined as one that is numerically well-formed and interpretable. For symbolic solvers, a valid answer is one derived within the time limit. We calculated ARR as the proportion of correct answers among all valid responses; the results are illustrated in Figure 11.

Our findings indicate that symbolic solvers (including InterGPS and our method) achieve substantially higher ARR than MLLMs, underscoring the inherent reliability of symbolic reasoning. Furthermore, our AutoGPS attains an ARR exceeding 90% across all experimental settings, outperforming InterGPS by 10.3% to 11.8%, which highlights the enhanced reliability of our framework. This improvement can be attributed to more rigorous formalization achieved by MPF, which reduce formalization inaccuracies and establish a more dependable foundation for symbolic reasoning.

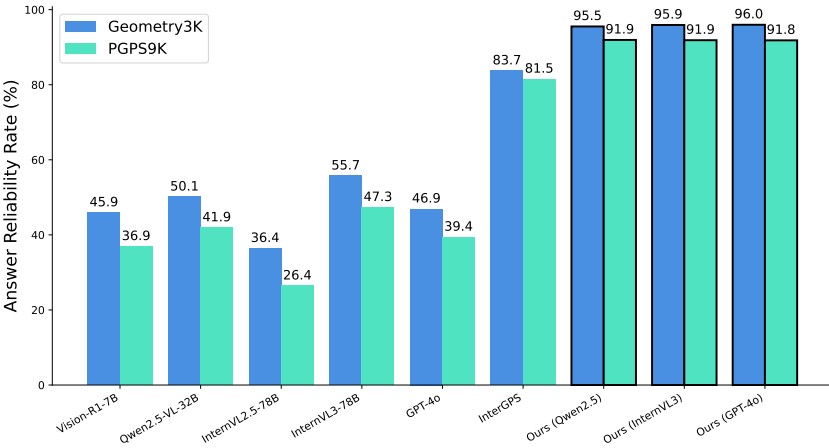

Figure 11: Answer reliability rate comparison among different methods.

**Additional Evaluation.**

To further demonstrate the generalization and scalability of proposed AutoGPS framework, we evaluated its performance on two additional datasets: MathVista (GPS) (Lu et al., 2023) and GeoQA (Chen et al., 2022b), as shown in Table 2 and Table 3. The proposed AutoGPS (InternVL3) achieves 86.2%, ranking 1st among existing open-source methods and competitive with several close-source models on MathVista (GPS) and achieves 86.2%, outperforming NGS, SCA-GPS, and DualGeoSolver, and is comparable to FGeo-HyperGNet (85.6%) on GeoQA. These results further validate the effectiveness and robustness of our AutoGPS framework in solving diverse geometry problems.

Table 2: MathVista (GPS) Performance Comparison

| Method | *Choice* **Acc.** |
|---|---|
| Human | 48.4 |
| *Close-Source* | |
| DreamPRM (o4-mini) (Cao et al., 2025) | 95.7 |
| Step R1-V-Mini | 89.9 |
| Kimi-k1.6-preview-20250308 | 91.8 |
| Doubao-pro-1.5 | 88.9 |
| *Open-Source* | |
| Vision-R1-7B (Huang et al., 2025b) | 82.7 |
| Ovis2_34B (Lu et al., 2025) | 84.6 |
| AutoGPS (InternVL3) | **86.2** |

Table 3: GeoQA Performance Comparison

| Method | *Choice* **Acc.** |
|---|---|
| Human (Chen et al., 2022b) | 92.3 |
| NGS (Chen et al., 2022b) | 57.4 |
| NGS-Auxiliary (Chen et al., 2022b) | 60.0 |
| SCA-GPS (Ning et al., 2023) | 64.1 |
| DualGeoSolver (Xiao et al., 2024) | 65.2 |
| FGeo-HyperGNet (Zhang et al., 2024c) | 85.6 |
| AutoGPS (InternVL3) | **86.2** |

**Performance Comparison of Symbolic Solvers.** We benchmarked the solving accuracy across different solvers using the Geometry3K ground-truth formalization. This approach effectively isolates the impact of formalization inaccuracies on problem-solving outcomes. As shown in Table 4, our approach achieves superior performance in most tasks, even surpassing human experts in average accuracy. This advantage of DSR originates from its enhanced capability for handling complex geometric configurations and an advanced symbolic reasoning algorithm with a larger theorem set.

Table 4: Symbolic solvers comparison using the Geometry3K ground-truth formalization by *Choice*.

| Method | Question Type | | | | Geometric Shape | | | | | Average |
|---|---|---|---|---|---|---|---|---|---|---|
| | *Angle* | *Length* | *Area* | *Ratio* | *Line* | *Triangle* | *Quad* | *Circle* | *Other* | |
| Human (Lu et al., 2021) | 53.7 | 59.3 | 57.7 | 42.9 | 46.7 | 53.8 | 68.7 | 61.7 | 58.3 | 56.9 |
| Human Expert (Lu et al., 2021) | 89.9 | 92.0 | 93.9 | 66.7 | 95.9 | 92.2 | 90.5 | 89.9 | 92.3 | 90.9 |
| InterGPS (Lu et al., 2021) | 83.1 | 77.9 | 62.3 | 75.0 | 86.4 | 83.3 | 77.6 | 61.5 | 70.4 | 78.3 |
| E-GPS (Wu et al., 2024) | 90.4 | 92.2 | 73.6 | **100.0** | 91.4 | 93.1 | 87.9 | 81.1 | 75.3 | 89.8 |
| GeoDRL (Peng et al., 2023) | 86.5 | 93.7 | 75.5 | **100.0** | 87.7 | 93.1 | 90.2 | 78.3 | 77.8 | 89.4 |
| HyperGNet(Zhang et al., 2024c) | - | - | - | - | - | - | - | - | - | 92.0 |
| Ours | **95.6** | **94.5** | **93.0** | 87.5 | **91.7** | **95.3** | **93.2** | **95.3** | **80.6** | **94.5** |

**Failed Cases Analysis.** In the human evaluation of reasoning processes, AutoGPS achieved 99% stepwise accuracy, with only one failed case. We provide this case in Figure 12. Although the final answer is correct, the formalization is not fully faithful to the original problem statement, violating the requirement for accurate comprehension of geometric information (Criterion 1 in Appendix A.2). The underlying causes are as follows:

(1) *Ambiguous Labeling*. The labels are spatially ambiguous. The label "13" could refer to either segment DC or CE, which may mislead both MLLMs and even human experts to incorrectly associate it with the nearer segment DC.

(2) *Training Corpus Bias*. The polygon "ABCD" appears far more frequently than "ABCE" in the LLM's training corpus, resulting in a bias, especially when a low decoding temperature is used.

Nevertheless, the formalization remains logically consistent and produces both sound reasoning steps and the correct answer. This case raises a more challenging question for future work: *How can we ensure faithful formalization that accurately reflects the original problem, even when the formalization is logically consistent and yields a correct answer?*

For MLLMs, we observed that stepwise accuracy was surprisingly higher than anticipated. We therefore further examined several failed cases and found that the correctly solved problems were generally simpler, requiring only straightforward reasoning without complex theorem applications or deep geometric understanding. Most complex problems, however, were not solved correctly, since MLLMs often failed to provide valid reasoning steps, leading to incorrect answers, and thus such cases were excluded from this category. Even when correct answers were produced, the following reasoning flaws were evident:

(1) Incorrect theorem application (e.g., Qwen2.5 in Figure 15).

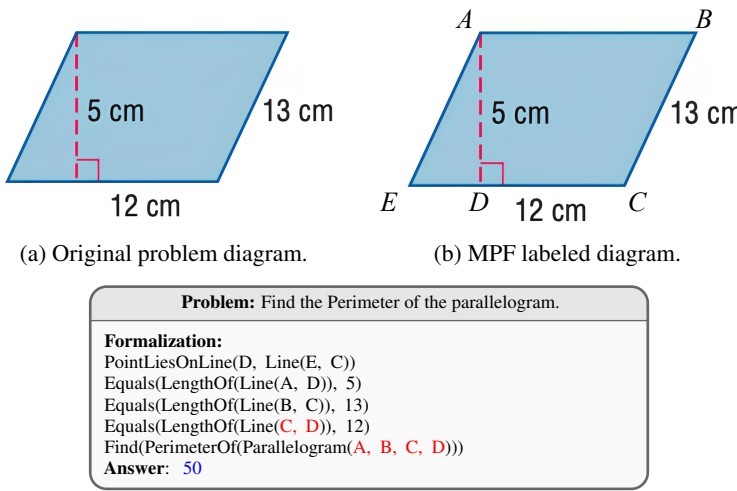

(a) Original problem diagram.    (b) MPF labeled diagram.

> **Problem:** Find the Perimeter of the parallelogram.
>
> **Formalization:**
> PointLiesOnLine(D, Line(E, C))
> Equals(LengthOf(Line(A, D)), 5)
> Equals(LengthOf(Line(B, C)), 13)
> Equals(LengthOf(Line(C, D)), 12)
> Find(PerimeterOf(Parallelogram(A, B, C, D)))
> **Answer**: 50

(c) Problem description and MPF formalization. The red part indicates the *unfaithful* formalization and the blue part indicates the correct answer.

Figure 12: Failed case of AutoGPS.

(2) Incorrect matching of geometric elements and text labels (e.g., GPT-4o in Figure 15).

(3) Incorrect understanding of geometric relationships (e.g., InternVL3 in Figure 15).

In most cases, MLLMs simultaneously suffered from multiple issues, occasionally arriving at correct conclusions while demonstrating incorrect outcomes across the majority of scenarios.

Notably, relatively few errors were found in the algebraic transformation process, which was surprising. These results provide us with some insights about how to further improve the geometric reasoning capability of MLLMs: (1) Enhancing the understanding of fine-grained geometric primitives and relationships, (2) Improving the theorem application capability.

**Solving Efficiency Analysis.** During the experiments, we set up a 1800-second timeout for the DSR to ensure high performance. In fact, most problems were solved in a much shorter time, as shown in Figure 13. For example, on the Geometry3K dataset with GPT-4o for the MPF module, the average solving time was only 57.59 seconds, with 99% of problems solved within 360 seconds (Figure 13a). It demonstrates the high efficiency of our DSR, which stems from high-quality formalization provided by the MPF and the efficient reasoning algorithm of the DSR.

**Ablation Study of Geometry Validation.** Geometry validation not only ensures the correctness of the generated formalization but also completes the implicit geometric relationships that are not explicitly stated in the diagram. We conducted an ablation study to evaluate the performance of our DSR with and without geometry validation, as shown in Table 5. Without geometry validation, the DSR was unable to fully capture the geometric relationships, even with the ground-truth formalization, reaching an accuracy ceiling around 56%. There are two reasons for this performance barrier: (1) Inconsistent formalization makes the problem unsolvable. (2) Many geometric relationships are implicit but important for problem solving. Since the *Geometry Validation* phase detects inconsistencies and supplements implicit geometric relationships, it provides the symbolic representation with error-free and complete geometric relationships, thereby significantly improving the performance of DSR.

**Ablation Study of Reasoning Strategies.** There are two reasoning strategies in our DSR: *Deductive Reasoning* (DR) and *Algebraic Reasoning* (AR). To evaluate their effectiveness, we conducted an ablation study by removing one of the two reasoning strategies and comparing the performance of our DSR using the Geometry3K ground-truth formalization, as shown in Table 6. When either reasoning strategy was removed, the DSR failed to solve most of the problems, where the accuracy dropped to 5.5% for both cases. Only when both reasoning strategies were employed did the DSR achieve a significant accuracy of 94.5%. DR's role is to derive new geometric relationships from existing ones through the application of geometric theorems, while AR's role is to solve algebraic equations and obtain values for unknown variables. Since geometry problems often require both

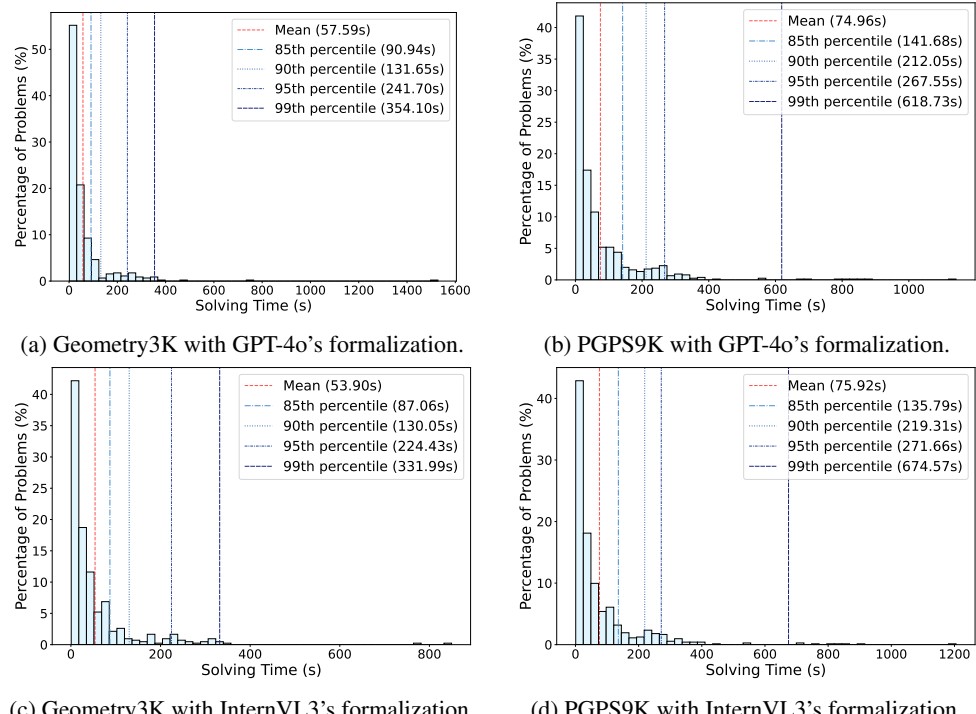

(a) Geometry3K with GPT-4o's formalization.      (b) PGPS9K with GPT-4o's formalization.

(c) Geometry3K with InternVL3's formalization.    (d) PGPS9K with InternVL3's formalization.

Figure 13: Solving time distribution of DSR.

theorem applications and algebraic transformations, the two reasoning strategies are complementary and indispensable, collaboratively enhancing the overall reasoning capability of the DSR.

**Ablation Study of Feedback and Refinement.** To rigorously evaluate the utility of our Feedback and Refinement mechanism, we conducted an ablation study comparing it with two standard multi-pass strategies: Pass@5 (a naive approach) and Major@5 (Self-Consistency). As shown in Table 7, the Refine@5 strategy demonstrates clear superiority in both efficacy and efficiency. Efficacy is evidenced by Refine@5 significantly outperforming Major@5 (Self-Consistency) by a substantial margin of $+4.9\%$ to $+7.0\%$ across all datasets (Geo3K, PGPS9K, and GeoQA). This finding strongly suggests that the directed refinement based on explicit feedback is vastly more effective for error correction than simply aggregating an undirected consensus from independent samples. Regarding Efficiency, Refine@5 achieves this superior performance with an average of only 1.36-1.51 total forward passes, dramatically lower than the 5 full, independent passes required by both Pass@5 and Major@5. This confirms that our mechanism not only yields higher accuracy but also does so in a computationally efficient manner.

Table 5: *Completion* accuracy of DSR with and without geometry validation on Geometry3K dataset.

| Geometry Validation | Formalization | |
|:---:|:---:|:---:|
| | MPF (GPT-4o) | Ground Truth |
| ✗ | 55.5 | 55.4 |
| ✔ | 75.4 | 94.5 |

Table 6: *Completion* accuracy of different reasoning strategies on Geometry3K dataset.

| | W/O DR | With DR |
|:---:|:---:|:---:|
| W/O AR | 0.0 | 5.5 |
| With AR | 5.5 | **94.5** |

Table 7: *Completion* accuracy of different forward strategies.

| Strategy | Geo3K | PGPS9K | GeoQA |
|----------|-------|--------|-------|
| Refine@5 | 71.2% | 73.3% | 71.9% |
| Pass@5 | 68.2% | 70.0% | 67.3% |
| Major@5 | 66.3% | 67.2% | 64.9% |

## C   LIMITATIONS AND FUTURE DIRECTIONS

- **Limited Geometric Formal Representation.** The formal language used in our work is expressive enough to represent most geometric configurations in the datasets. However, it is not sufficient to represent some corner cases, such as three tangential circles. General-purpose formal languages such as Lean (De Moura et al., 2015; Moura & Ullrich, 2021) and Coq (Bertot & Castéran, 2013) still require a large amount of groundwork to describe the geometry problems at present. Formalization with those languages requires deep expertise in both geometry and formal languages, which is a high barrier for most researchers. This challenge complicates the annotation of large-scale geometry datasets. Additionally, due to the limited existing training data, LLMs are not able to understand such complex formal languages well, resulting in low performance (Wu et al., 2022; Li et al., 2024a). In this paper, we do not pursue a complete solution to geometry representation, as it is a separate and extremely challenging research topic that demands substantial investment from the mathematical formalization community.

- **Large-Scale Annotated Geometry Datasets for AI Community.** The creation of large-scale geometry datasets remains challenging due to the labor-intensive nature of manual collection and annotation(Huang et al., 2024). Current benchmarks like Geometry3K (Lu et al., 2021) and PGPS9K (Zhang et al., 2023) not only suffer from limited scale but also lack detailed step-by-step solutions. Our AutoGPS addresses these limitations by enabling the automated generation of high-quality formalizations and rigorous reasoning steps, which can be utilized to create large-scale datasets with comprehensive solution documentation. This will empower the community to develop more robust neural models with enhanced reasoning fidelity and reduced hallucination risks.

- **Trade-off between Proof Conciseness and Computational Efficiency.** The proposed methodology employs a deductive reasoning-based algorithm that operates by exhaustively searching and applying all potential axioms to progressively expand the reasoning hypergraph, aiming to enable backtracking to identify the shortest proof path. The imperative to find the shortest proof requires exhaustive exploration of various proof approaches to find the optimal path, which fundamentally limits algorithmic efficiency through combinatorial explosion. Should computational efficiency be prioritized over proof conciseness, heuristic search strategies would be a superior alternative, though at the expense of guaranteeing solution optimality. This fundamental tension reveals an inherent trade-off between the dual goals of minimal proof determination and highly efficient problem solving, requiring us to find a balance between the two objectives in future work.

# D SOLUTION EXAMPLES

## D.1 STEPWISE REASONING PROCESS FOR NOISE DATA

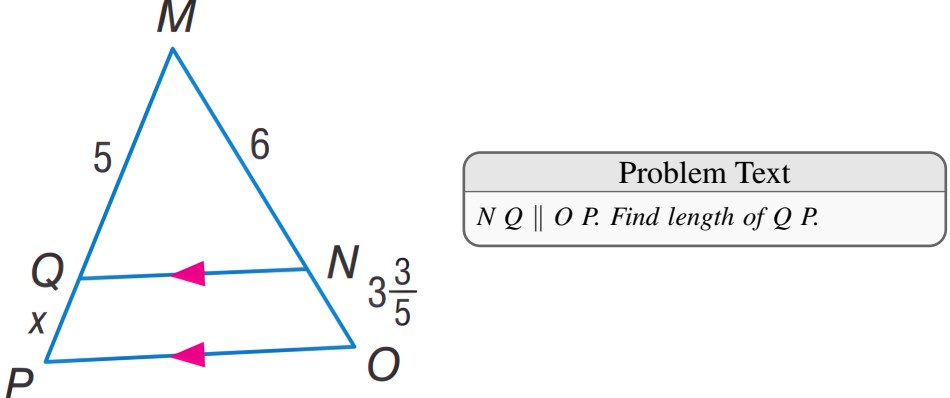

**Problem Text**

$N\,Q \parallel O\,P$. Find length of $Q\,P$.

### Solution by AutoGPS

**Step 1** : Known facts : $start \implies x = \overline{PQ}$, $6 = \overline{MN}$, $3 + \dfrac{3}{5} = \overline{NO}$, $Q\,on\,\overline{MP}$, $\angle NMP, \angle OMP$

     $\overline{NQ} \parallel \overline{OP}$, $5 = \overline{MQ}$, $N\,on\,\overline{MO}$

**Step 2** : Line Segment Split : $Q\,on\,\overline{MP} \implies \overline{MP} = \overline{MQ} + \overline{PQ}$

**Step 3** : Same Angle : $\angle NMP, Q\,on\,\overline{MP} \implies \angle NMP = \angle NMQ$

**Step 4** : Corresponding Angle Theorem : $\overline{NQ} \parallel \overline{OP} \implies \angle MNQ = \angle MOP, \angle MPO = \angle MQN$

**Step 5** : Line Segment Split : $N\,on\,\overline{MO} \implies \overline{MO} = \overline{MN} + \overline{NO}$

**Step 6** : Same Angle : $\angle OMP, N\,on\,\overline{MO} \implies \angle NMP = \angle OMP$

**Step 7** : Substitution : $\overline{MP} = \overline{MQ} + \overline{PQ}$, $x = \overline{PQ}$, $5 = \overline{MQ} \implies 5 + x = \overline{MP}$

**Step 8** : Substitution : $6 = \overline{MN}$, $\overline{MO} = \overline{MN} + \overline{NO}$, $3 + \dfrac{3}{5} = \overline{NO} \implies 6 + (3 + \dfrac{3}{5}) = \overline{MO}$

**Step 9** : Transtivity of Equivalence : $\angle NMP = \angle OMP, \angle NMP = \angle NMQ \implies \angle NMQ = \angle OMP$

**Step 10** : Solve Linear Equation System : $6 + (3 + \dfrac{3}{5}) = \overline{MO} \implies 9.6 = \overline{MO}$

**Step 11** : Angle-Angle Similarity Theorem : $\angle NMQ = \angle OMP, \angle MNQ = \angle MOP, \angle MPO = \angle MQN$

     $\implies \triangle MNQ \sim \triangle MOP$

**Step 12** : Similar Definition : $\triangle MNQ \sim \triangle MOP \implies \angle NMQ = \angle OMP, sim\_ratio = \dfrac{\overline{MN}}{\overline{MO}}$

     $sim\_ratio = \dfrac{\overline{MQ}}{\overline{MP}}, \angle MNQ = \angle MOP$

**Step 13** : Substitution : $6 = \overline{MN}$, $9.6 = \overline{MO}$, $sim\_ratio = \dfrac{\overline{MN}}{\overline{MO}} \implies \dfrac{6}{9.6} = sim\_ratio$

**Step 14** : Substitution : $sim\_ratio = \dfrac{\overline{MQ}}{\overline{MP}}$, $5 + x = \overline{MP}$, $5 = \overline{MQ} \implies sim\_ratio = \dfrac{5}{(5 + x)}$

**Step 15** : Transtivity of Equivalence : $sim\_ratio = \dfrac{5}{(5 + x)}, \dfrac{6}{9.6} = sim\_ratio \implies \dfrac{6}{9.6} = \dfrac{5}{(5 + x)}$

**Step 16** : Solve Linear Equation System : $\dfrac{6}{9.6} = \dfrac{5}{(5 + x)}, x = \overline{PQ} \implies 3 = \overline{PQ}$

Figure 14: Reasoning process of AutoGPS for a noise problem.

## D.2 REASONING PROCESSES COMPARISON

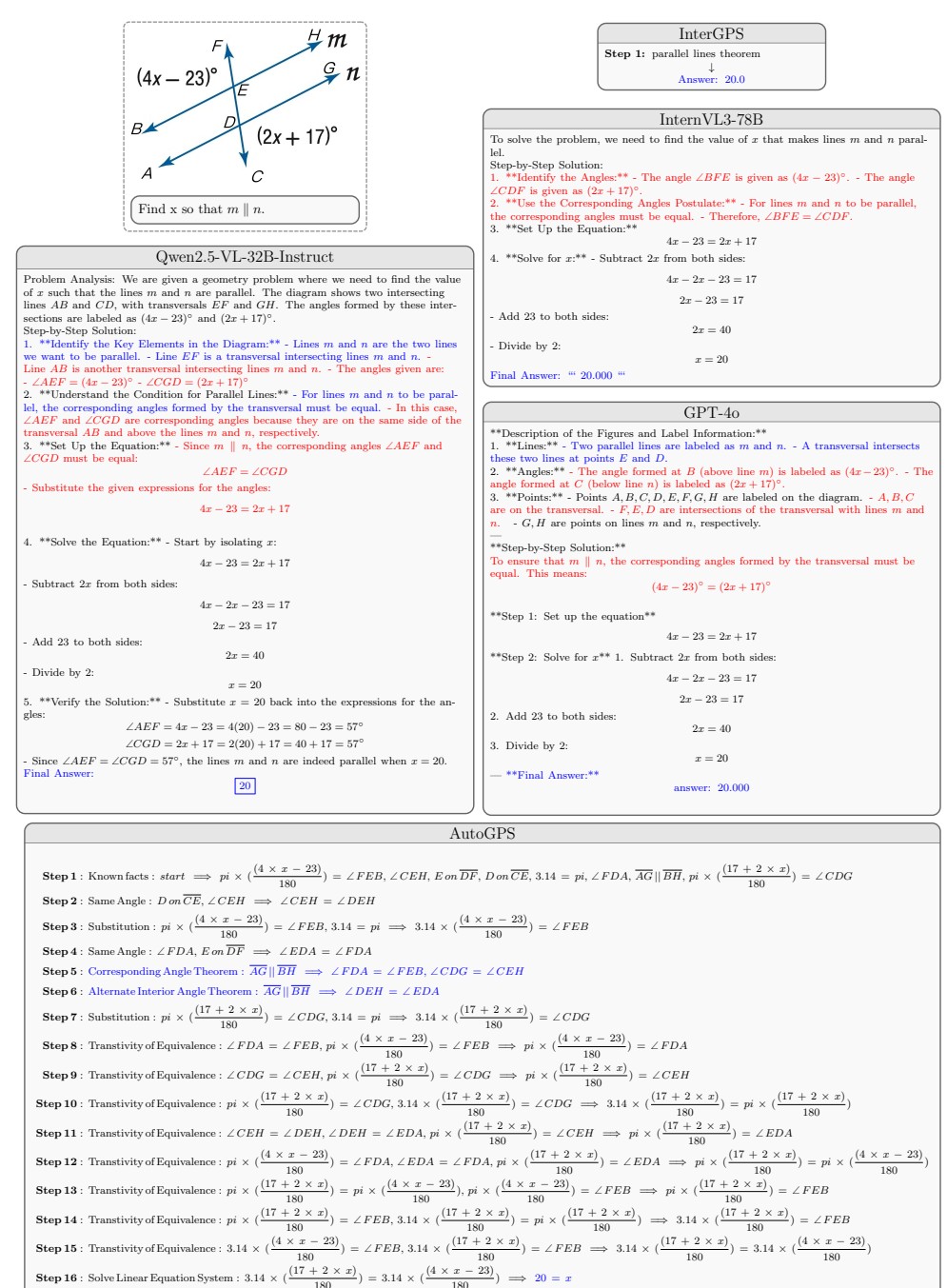

Figure 15: Comparison of reasoning processes (1). All three MLLMs generated **correct answer** but provided logically **incorrect reasoning process**. Symbolic-based method, InterGPS, provided the theorem applied but **lacked detailed reasoning process**. In constract, AutoGPS generated **logically coherent stepwise solution** with correct answer.

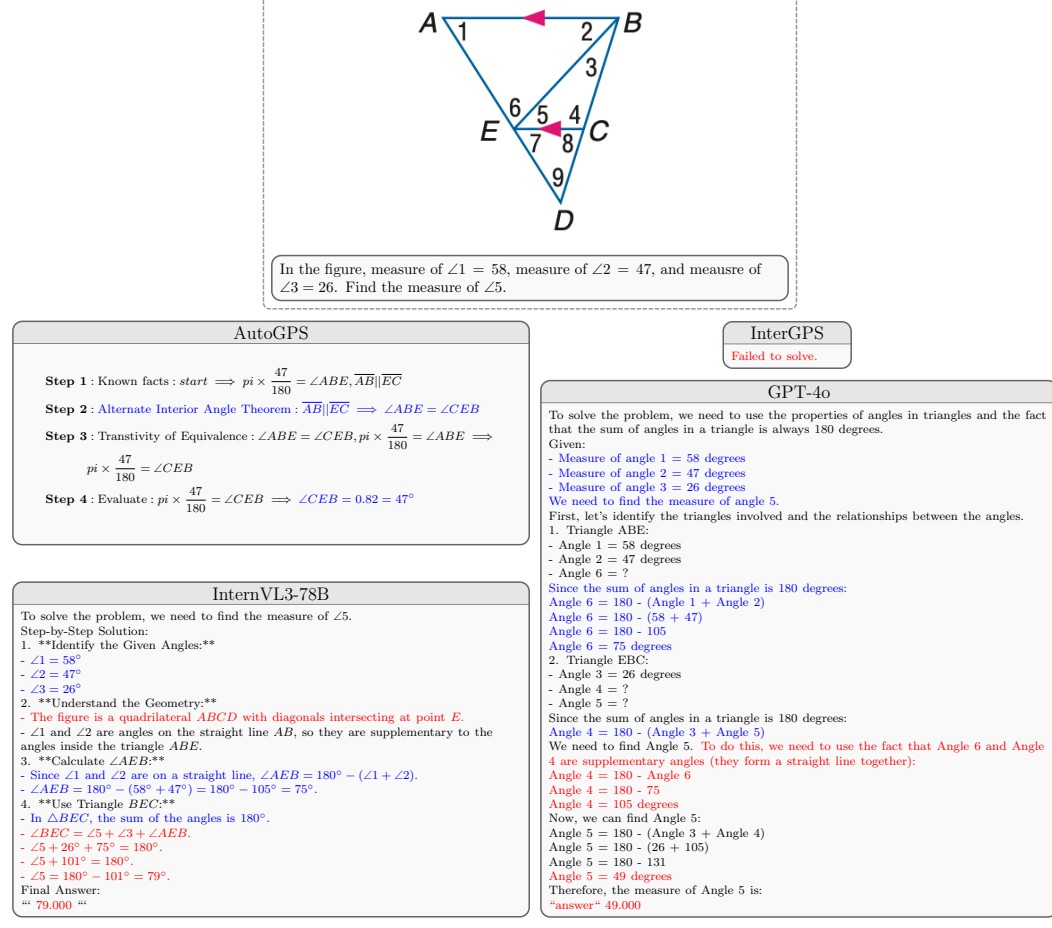

Figure 16: Comparison of reasoning processes (2). MLLMs captured the **correct geometric information** but generated **wrong reasoning processes, leading to incorrect answers**. Symbolic-based method, InterGPS, failed to solve this problem. AutoGPS generated **a concise stepwise solution** with correct answer.

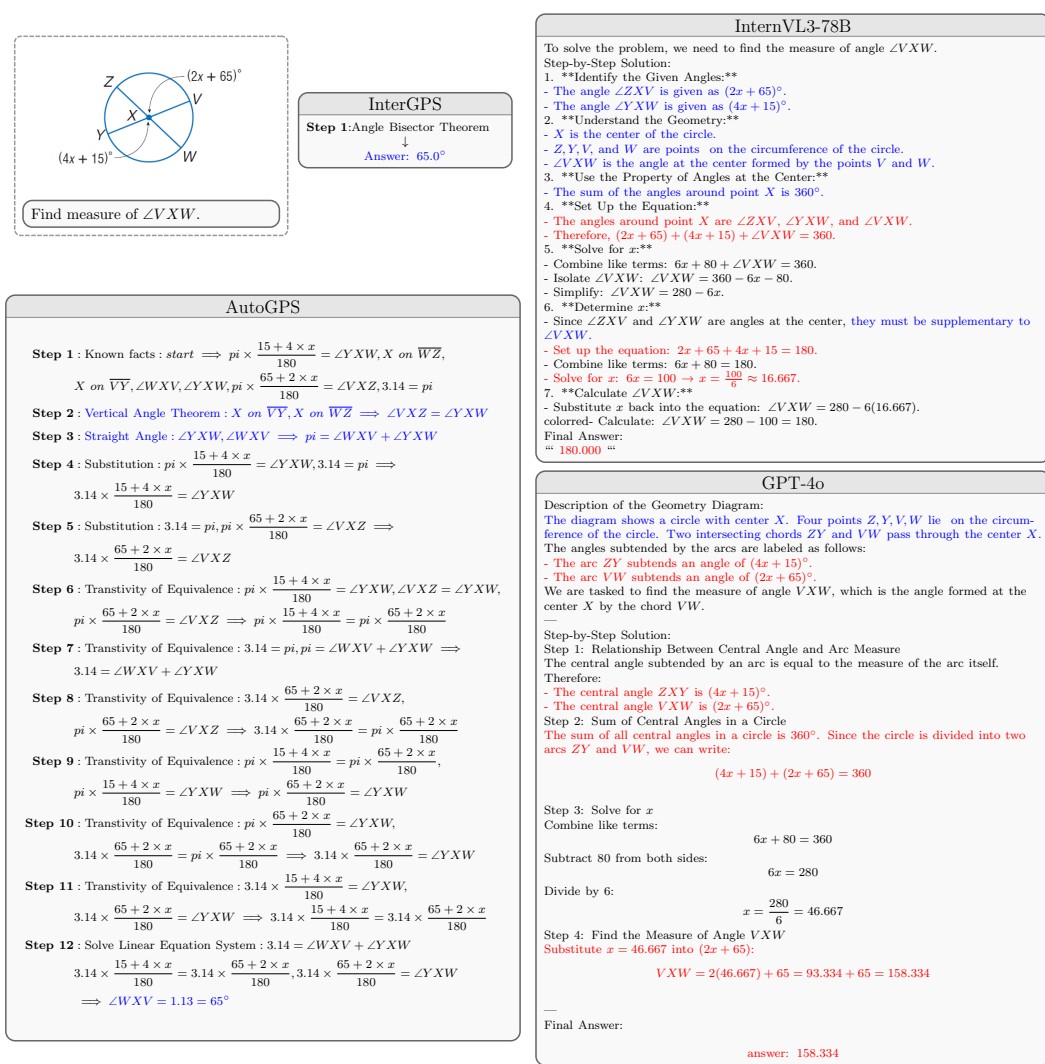

Figure 17: Comparison of reasoning processes (3). MLLMs **failed to capture correct geometric relationships** and generated **wrong reasoning processes**. Symbolic-based method, InterGPS, provided correct answer but **lacked detailed reasoning process**. leading to the wrong answers. AutoGPS generated a **human-readable reasoning process** with correct answer.

# E    HYPERGRAPH EXAMPLES

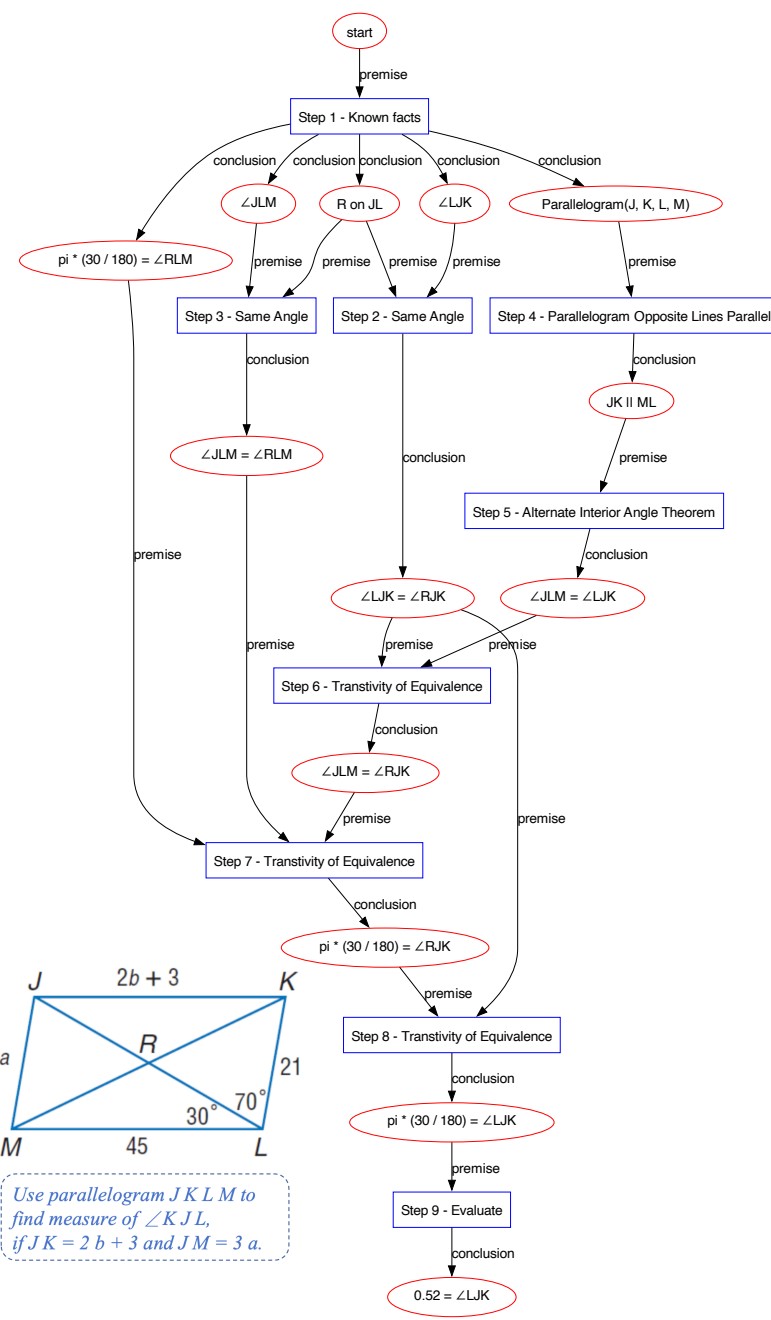

Figure 18: Minimal reasoning hypergraph example (1). The red nodes indicate literals. The blue boxes indicate derivation hyperedges connecting the premise nodes to the conclusion nodes. The hyperedges are topologically sorted to indicate the order of reasoning.

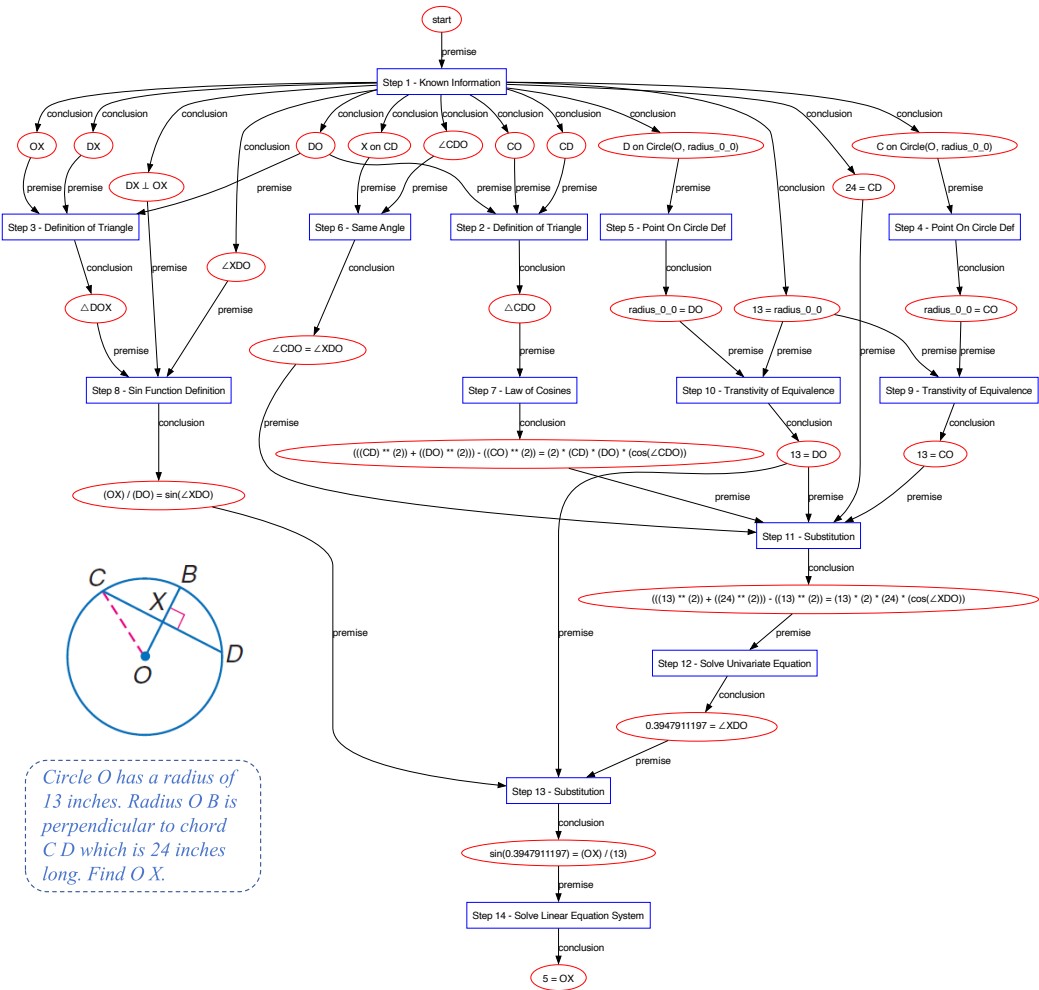

Figure 19: Minimal reasoning hypergraph example (2). The red nodes indicate literals. The blue boxes indicate derivation hyperedges connecting the premise nodes to the conclusion nodes. The hyperedges are topologically sorted to indicate the order of reasoning.

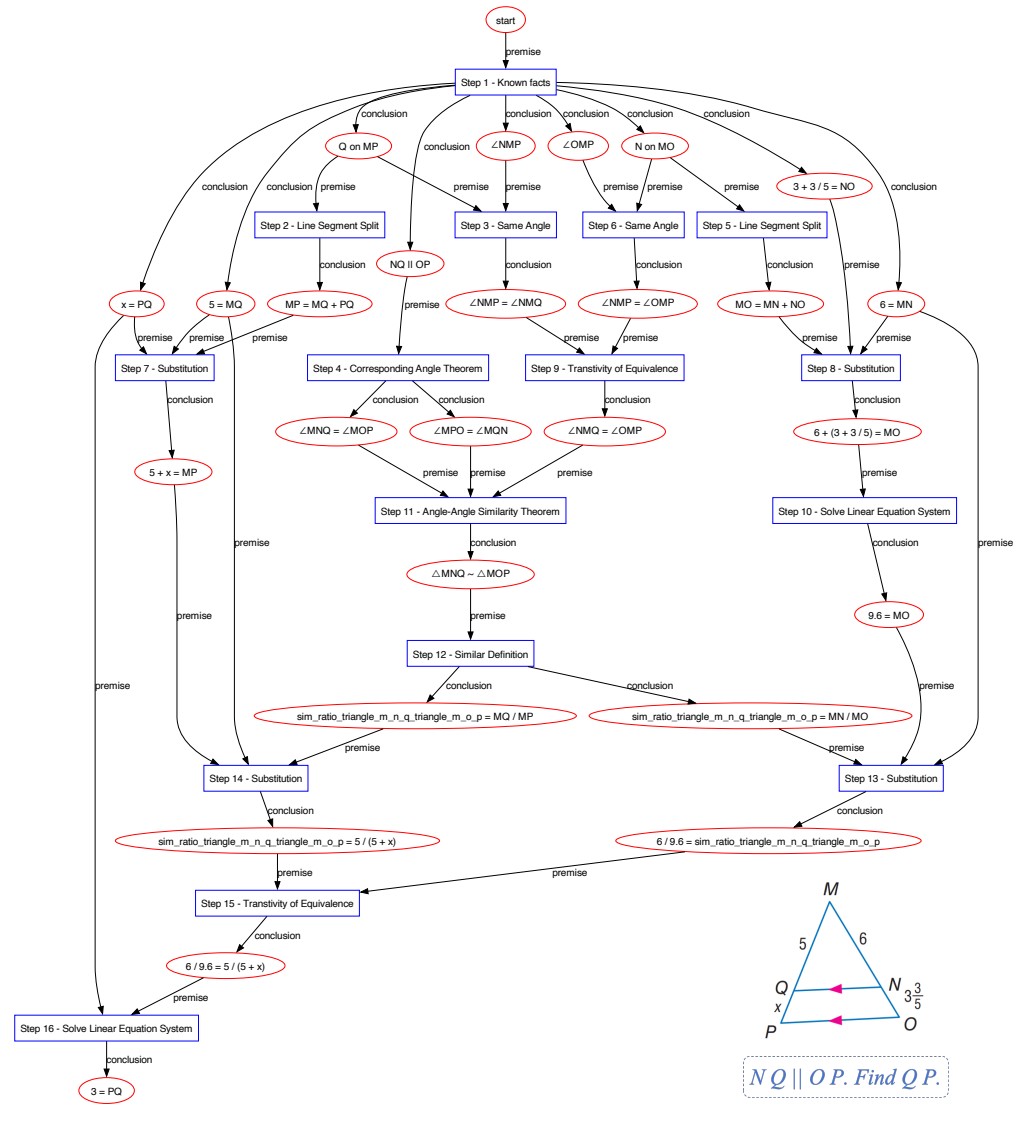

Figure 20: Minimal reasoning hypergraph example (3). The red nodes indicate literals. The blue boxes indicate derivation hyperedges connecting the premise nodes to the conclusion nodes. The hyperedges are topologically sorted to indicate the order of reasoning.

# F  FORMAL LANGUAGE DEFINITIONS

We have appropriately simplified the formal language used in InterGPS (Lu et al., 2021), and described its syntax via a context-free grammar (CFG). This mathematical definition facilitates efficient parsing of the formal language. Let the special character set be $\texttt{SPECIAL\_CHAR} = \{\_, \backslash, (, ), , , , +, -, *, /, ., , \{, \}, \text{`}, \$, \text{'}\}$. The grammar is defined as a quadruple $G = (N, \Sigma, P, S)$, where:

$$N = \{\text{logic\_form}, \text{args}, \text{arg}, \text{id}, \text{expr}\}$$
$$\Sigma = \texttt{UPPER\_LETTER} \cup \texttt{ALPHA\_NUM} \cup \texttt{SPECIAL\_CHAR}$$
$$S = \text{logic\_form}$$
$$P :$$
$$\text{logic\_form} \rightarrow \text{id} \, (\, \text{args} \, ) \mid \text{id} \mid \text{expr}$$
$$\text{args} \rightarrow \text{arg} \, (, \, \text{arg})^*$$
$$\text{arg} \rightarrow \text{logic\_form} \mid \text{id} \mid \text{expr}$$
$$\text{id} \rightarrow \texttt{UPPER\_LETTER} \, (\texttt{ALPHA\_NUM} \cup \{\_\})^*$$
$$\text{expr} \rightarrow [-] \, \texttt{ANUB} \, (\texttt{ALPHA\_NUM} \cup \texttt{SPECIAL\_CHAR})^*$$
$$\texttt{UPPER\_LETTER} = \{\texttt{A}, \texttt{B}, \ldots, \texttt{Z}\}$$
$$\texttt{ALPHA\_NUM} = \{\texttt{a}, \ldots, \texttt{z}\} \cup \{\texttt{A}, \ldots, \texttt{Z}\} \cup \{\texttt{0}, \ldots, \texttt{9}\}$$
$$\texttt{ANUB} = \texttt{ALPHA\_NUM} \cup \{\_, \backslash\}$$

A detailed illustrative example of the formal language is provided below:

Table 8: Predicate and literal definitions for the formal language (1).

| Literals | Explanation |
|---|---|
| Line(A,B) | A line segment with endpoints A and B |
| Angle(A) | The angle with point A as vertex |
| Angle(A,B,C) | Angle ABC with B as the vertex |
| Triangle(A,B,C) | Triangle with vertices A, B, and C |
| Quadrilateral(A,B,C,D) | Quadrilateral with vertices A, B, C, and D |
| Parallelogram(A,B,C,D) | Parallelogram with vertices A, B, C, and D |
| Square(A,B,C,D) | Square with vertices A, B, C, and D |
| Rectangle(A,B,C,D) | Rectangle with vertices A, B, C, and D |
| Rhombus(A,B,C,D) | Rhombus with vertices A, B, C, and D |
| Trapezoid(A,B,C,D) | Trapezoid with vertices A, B, C, and D |
| Kite(A,B,C,D) | Kite with vertices A, B, C, and D |
| Polygon(A,B,C,...) | Polygon with vertices A, B, C, etc. |
| Pentagon(A,B,C,D,E) | Pentagon with vertices A, B, C, D, and E |
| Hexagon(A,B,C,D,E,F) | Hexagon with vertices A, B, C, D, E, and F |
| Heptagon(A,B,C,D,E,F,G) | Heptagon with vertices A, B, C, D, E, F, and G |
| Octagon(A,B,C,D,E,F,G,H) | Octagon with vertices A, B, C, D, E, F, G, and H |
| Circle(A) | Circle with center A |
| Circle(O, r) | Circle with center O and radius r |
| Arc(A,B) | Minor arc with A and B as endpoints on circle |
| Arc(A,B,C) | Arc that passes through points A, B, and C |
| Sector(O,A,B) | Sector of a circle with center O and points A and B on the circumference |

Table 9: Predicate and literal definitions for the formal language (2).

| Literals | Explanation |
|---|---|
| Equilateral(Polygon(A,B,C,D)) | Polygon ABCD is equilateral |
| Regular(Polygon(A,B,C,D)) | Polygon ABCD is regular |
| AreaOf(Shape(...)) | Area of the Shape ... |
| PerimeterOf(Shape(...)) | Perimeter of the Shape ... |
| RadiusOf(Circle(O)) | Radius of the circle O |
| DiameterOf(Circle(O)) | Diameter of the circle O |
| CircumferenceOf(Circle(O)) | Circumference of the circle O |
| MeasureOf(Angle(A, B, C)) | Measure of the angle ABC |
| MeasureOf(Arc(A, B)) | Measure of the arc AB |
| LengthOf(Line(A, B)) | Length of the line segment AB |
| PointLiesOnLine(A,Line(B,C)) | Point A lies on segment BC |
| PointLiesOnCircle(A,Circle(O,r)) | Point A lies on the circle with center O and radius r |
| Parallel(Line(A,B),Line(C,D)) | Line AB is parallel to Line CD |
| Perpendicular(Line(A,B),Line(C,D)) | Line AB is perpendicular to Line CD |
| BisectsAngle(Line(A,B),Angle(X,A,Y)) | Line AB bisects angle XAY |
| Congruent(Triangle(A,B,C),Triangle(D,E,F)) | Triangle ABC is congruent to triangle DEF |
| Similar(Triangle(A,B,C),Triangle(D,E,F)) | Triangle ABC is similar to triangle DEF |
| Tangent(Line(A,B),Circle(O,r)) | Line AB is tangent to circle O with radius r |
| Secant(Line(A,B),Circle(O,r)) | Line AB is a secant to circle O with radius r |
| CircumscribedTo(Shape(...),Shape(...)) | First shape is circumscribed to the second shape |
| InscribedIn(Shape(...),Shape(...)) | First shape is inscribed in the second shape |
| IsMidpointOf(C,Line(A,B)) | Point C is the midpoint of line AB |
| IsCentroidOf(O,Triangle(A,B,C)) | Point O is the centroid of triangle ABC |
| IsIncenterOf(O,Triangle(A,B,C)) | Point O is the incenter of triangle ABC |
| IsRadiusOf(Line(O,A),Circle(O,r)) | Line OA is a radius of circle O with radius r |
| IsDiameterOf(Line(A,B),Circle(O,r)) | Line AB is a diameter of circle O with radius r |
| IsMidsegmentOf(Line(A,B),Triangle(D,E,F)) | Line AB is a midsegment of triangle DEF |
| IsChordOf(Line(A,B),Circle(O,r)) | Line AB is a chord of circle O with radius r |
| IsPerpendicularBisectorOf(Line(A,B),Line(C,D)) | Line AB is the perpendicular bisector of line CD |
| IsMedianOf(Line(E,F),Trapezoid(A,B,C,D)) | Line EF is the median of trapezoid ABCD |
| IsMedianOf(Line(E,F),Triangle(A,B,C)) | Line EF is a median of triangle ABC |
| SinOf(var) | Sine of var (var can be variable, measure of angle or arc) |
| CosOf(var) | Cosine of var (var can be variable, measure of angle or arc) |
| TanOf(var) | Tangent of var (var can be variable, measure of angle or arc) |
| CotOf(var) | Cotangent of var (var can be variable, measure of angle or arc) |
| HalfOf(var) | Half of var (var can be variable, length, measure, area, etc.) |
| SqrtOf(var) | Square root of var (var can be variable, length, measure, area, etc.) |
| RatioOf(var1,var2) | Ratio of var1 to var2 (can be variables, lengths, measures, areas, etc.) |
| Add(var1,var2,...) | Addition of var1, var2, and possibly more variables |
| Mul(var1,var2,...) | Multiplication of var1, var2, and possibly more variables |
| Sub(var1,var2) | Subtraction of var2 from var1 |
| Div(var1,var2) | Division of var1 by var2 |
| Pow(var1,var2) | var1 raised to the power of var2 |
| Equals(var1,var2) | var1 equals var2 (a = b is equivalent to Equals(a, b)) |
| Find(var) | Find the value of the variable |

## G  PROMPT TEMPLATE

In our experiments, we involve the use of multimodal large language models to accomplish the following tasks:

- *Choice* Task. Solving multiple-choice questions.

- *Completion* Task. Solving fill-in-the-blank questions.

- *Formalization* Task. Direct formalization of problems without pre-formalization.

- *Alignment* Task. Aligning the results with pre-formalization specifications.

The corresponding prompt templates for each task are presented below. The image input is omitted for brevity. The variables are colored in red to indicate the need for replacement with the corresponding content.

---

**Prompt Template for *Choice* Task**

This is a geometry problem. The problem text is given as " {problem text} "
There are several choices:
- Choices are: A. {content_A}, B. {content_B}, C. {content_C}, D. {content_D}

**Tasks:**
- Describe the figures and label information in the geometry diagram.
- Solve the problem step by step and give your final choice in the form of "choice". If your choice is A, give "A" at last.

---

**Prompt Template for *Formalization* Task**

Given the geometry problem with problem text "'{problem text}'", we use logic forms to describe the information of this problem. The logic forms are defined as follows:
"'plaintext

```
[
### Predicate Definitions
\{predicate\_definition\}
]
```
"'
**Task:**
- Identify the geometric figures in the diagram and list the known value information in the diagram.
- Formalize the problem with the given logic forms. Give your final logic forms in one single plaintext code block.

**Note:**
- A line named t with endpoints A and B, then it is expressed as Line(A, B) rather than Line(t).
- A circle with center O with a radius of 5 is expressed as Circle(O, radius_o) and Equals(radius_o, 5).
- A line segment with length 10 is expressed as Equals(LengthOf(Line(A, B)), 10).
- An angle ABC with measure 30 degrees is expressed as Equals(MeasureOf(Angle(A, B, C)), 30).
- An arc AB with measure 60 degrees is expressed as Equals(MeasureOf(Arc(A, B)), 60).
- A point A lies on segment BC is expressed as PointLiesOnLine(A, Line(B, C)).
- A point A lies on circle with center O and radius r is expressed as PointLiesOnCircle(A, Circle(O, r)).
- If the goal is to find the area of a shaded region, use the arithmetic operation expression with other regular figures to represent the shaded region. For example, Sub(AreaOf(Circle(C)), AreaOf(Triangle(D, E, F))).
- Each problem should have a goal in the form of Find(...), for example, Find(LengthOf(Line(X, Y))).
- Please formalize the problem faithfully. Do not add any extra information or do deduction.

---

---

**Prompt Template for *Completion* Task**

This is a geometry problem. The problem text is given as " {problem text} "

**Tasks:**
- Describe the figures and label information in the geometry diagram.
- Solve the problem step by step and give the final answer in the form of "answer" and round it to 3rd decimals. If the answer is 5, give "5.000" at last.

---

**Prompt Template for *Alignment* Task**

Given the geometry problem with problem text "'{problem text}'", we use logic forms to describe the information of this problem. The logic forms are defined as follows:
'''plaintext

```
[
### Predicate Definitions
\{predicate\_definition\}
]
```
'''
We have previously parsed the diagram and text. The diagram logic forms are:
'''plaintext

```
[
\{diagram_logic_form_1\}
\{diagram_logic_form_2\}
...
]
```
'''
And the text logic forms are:
'''plaintext

```
[
\{text_logic_form_1\}
\{text_logic_form_2\}
...
]
```
'''
**Task:**
- Replace $ with point identifier and replace Shape($) with specific geometric figures.
- Check if the problem is correctly converted to logic forms.
- Combine the final diagram logic forms and text logic forms in the format of plain text code block.

**Note:**
- A circle with center O with a radius of 5 is expressed as Circle(O, radius) and Equals(RadiusOf(Circle(O)), 5).
- A circle with center O with a diameter of 10 is expressed as Circle(O, radius) and Equals(DiameterOf(Circle(O)), 10).
- A line segment with length 10 is expressed as Equals(LengthOf(Line(A, B)), 10).
- An angle ABC with measure 30 degrees is expressed as Equals(MeasureOf(Angle(A, B, C)), 30).
- An arc AB with measure 60 degrees is expressed as Equals(MeasureOf(Arc(A, B)), 60).
- If the goal is to find the area of a shaded region, use the arithmetic operation expression with other regular figures to represent the shaded region. For example, Sub(AreaOf(Circle(C)), AreaOf(Triangle(D, E, F))).

