# OpenReview forum: "AutoGPS: Automated Geometry Problem Solving via Multimodal Formalization and Deductive Reasoning"
_ICLR.cc/2026/Conference — ICLR 2026 Poster_

### Official Review · Reviewer_JksY · 2025-10-27

**Soundness:** 4
**Presentation:** 2
**Contribution:** 3
**Rating:** 6
**Confidence:** 5

**Summary:**

This paper introduces a novel neuro-symbolic framework for GPS problems, featuring two key components: the Multimodal Problem Formalizer (MPF) and the Deductive Symbolic Reasoner (DSR). These components work collaboratively to address the challenges inherent in GPS problems. Experimental results demonstrate that the proposed method yields very promising outcomes, indicating significant potential in this area of research.

**Strengths:**

1. This paper is technically sound. The authors introduce two novel modules, the Multimodal Problem Formalizer (MPF) and the Deductive Symbolic Reasoner (DSR), both of which require substantial human effort to develop. These components have the potential to greatly benefit the GPS research community.
2. The experiments presented in this paper are both promising and comprehensive. The proposed method can be integrated with various base MLLMs, significantly improving their performance, especially in completion tasks.
3. Overall, the paper is well-written and easy to follow.

**Weaknesses:**

1. The authors state that MPF and DSR collaborate to formalize the original GPS problem. However, it is unclear whether this process may lead to incorrect problem formulations. The paper would benefit from a discussion and analysis of both successful and failed cases, including a reported success rate for the formalization process. Such information would be valuable for subsequent research to better understand the strengths and limitations of the proposed approach.
2. The paper does not report the API or inference costs associated with each method. As neuro-symbolic approaches often introduce additional inference and interaction steps when addressing GPS problems, it is important to assess whether these extra costs are justified by the observed performance improvements. Additionally, it would be worthwhile to investigate whether simpler techniques, such as self-consistency, could achieve similar performance gains within comparable evaluation budgets.
3. The discussion of related work could be expanded. In particular, the paper should cite and analyze earlier neuro-symbolic methods for mathematical reasoning problems, such as [1], [2], [3], and [4]. This would help position the proposed approach within the broader context of existing research.

**Reference**

[1] Zenan Li, Zhi Zhou, Yuan Yao, Xian Zhang, Yu-Feng Li, Chun Cao, Fan Yang, Xiaoxing Ma. **Neuro-Symbolic Data Generation for Math Reasoning.** NeurIPS 2024.

[2] Ning Shang, Yifei Liu, Yi Zhu, Li Lyna Zhang, Weijiang Xu, Xinyu Guan, Buze Zhang, Bingcheng Dong, Xudong Zhou, Bowen Zhang, Ying Xin, Ziming Miao, Scarlett Li, Fan Yang, Mao Yang. **rStar2-Agent: Agentic Reasoning Technical Report.** Arxiv 2025.

[3] Zenan Li, Zhaoyu Li, Wen Tang, Xian Zhang, Yuan Yao, Xujie Si, Fan Yang, Kaiyu Yang, Xiaoxing Ma. **Proving Olympiad Inequalities by Synergizing LLMs and Symbolic Reasoning.** ICLR 2025

[4] Weiming Wu, Zi-kang Wang, Jin Ye, Zhi Zhou, Yu-Feng Li, Lan-Zhe Guo. **NeSyGeo: A Neuro-Symbolic Framework for Multimodal Geometric Reasoning Data Generation.** Arxiv 2025.

**Questions:**

Please refer to the `Weakness` section.

---

> ### Author Response · Authors · 2025-11-25
>
> We thank the reviewer for the constructive feedback. We address the concerns as follows:
>
> ## R1. Formalization Correctness
>
> The success rate of our formalization process is quantitatively reflected in several ways:
>
> (i) **Direct Quality Evaluation:** It is captured by the evaluation of formalization quality (Figure 5 in the main paper).
>
> (ii) **Answer Reliability Analysis:** Crucially, the formalization is considered a success only if it leads to the correct solution. In Appendix B (**Answer Reliability Analysis**), we report the **solution reliability** (i.e., the ratio of _solved correctly_ to _solved_), which serves as a reliable proxy for the formalization success rate. Our method achieves an impressive **over 90% success rate** in formalization based on this metric.
>
> We have included one insightful failure case in the appendix (**Failed Cases Analysis**). Acknowledging the reviewer's suggestion that more examples would be valuable, we commit to adding **additional meaningful failure cases** in the final version to provide a more comprehensive qualitative analysis of the formalization process's strengths and limitations.
>
> ## R2. Inference Costs: Efficiency and Comparison with Self-Consistency
>
> We thank the reviewer for raising the important point of API and inference costs. While we did not calculate the explicit monetary cost, we provide a robust analysis to demonstrate that our AutoGPS approach is highly **compute-efficient** and **superior to the Self-Consistency (SC) technique**.
>
> ### 1. Minimal Overhead of AutoGPS
>
> Our core innovations, **Multimodal Alignment** and the **Feedback & Refinement Phase**, were designed for computational efficiency:
>
> (i) **Low Average Cost:** The **Refinement@5** mechanism, where a single solution attempt is limited with a maximum of five iterations, requires only **1.36–1.51 forward passes on average**. This is a marginal increase over a single-pass inference.
>
> (ii) **Direct and Concise Output:** Crucially, we achieve this without the need for additional, computationally expensive reasoning steps (like standard Chain-of-Thought), thus maintaining minimal overhead and justifying the performance gain.
>
> ### 2. Efficiency and Efficacy: Refinement vs. Self-Consistency
>
> The reviewer asked for a comparison against simplier techniques like *Self-Consistency*. We explicitly compared our Refinement mechanism with two strategies that utilize multiple forward passes: the naive **Pass@5** and the **Major@5 (Majority Voting)**, which is the standard implementation of the Self-Consistency technique.
>
> The results, based on *strict Completion Accuracy*, demonstrate that Refinement is both more effective and more efficient:
>
> | Strategy | Description | Geo3K | PGPS9K | GeoQA |
> | :--- | :--- | :--- | :--- | :--- |
> | **Refine@5** | Single Attempt, $\le 5$ Feedback Iterations | **71.2%** | **73.3%** | **71.9%** |
> | Pass@5 | 5 Independent Forward Passes (Pass@K) | 68.2% | 70.0% | 67.3% |
> | **Major@5** | 5 Independent Forward Passes with **Majority Voting (Self-Consistency)** | 66.3% | 67.2% | 64.9% |
>
> **(i) Efficacy:** **Refine@5** significantly outperforms **Major@5 (Self-Consistency)** by **+4.9% to +7.0%** across all datasets. This indicates that **directed refinement based on feedback** is vastly more effective than **undirected consensus gathering** from independent samples.
>
> **(ii) Efficiency:** **Refine@5** achieves this superior performance while requiring an average of only **1.36–1.51 total forward passes**. In contrast, both **Pass@5** and the **Major@5 (Self-Consistency)** strategy require **5 full, independent forward passes**.
>
> This comparative analysis confirms that the AutoGPS refinement mechanism is **effective** and **compute-efficient**.
>
>
> ## R3. Related Work Extension
>
> We acknowledge that our discussion of prior neuro-symbolic work was limited. We will **significantly expand the Related Work section with a new subsection titled “Neural-Symbolic Methods for Mathematics”** in the revised manuscript to include a thorough analysis of the suggested references ([1–4]), properly positioning our approach within the broader context of recent neuro-symbolic methods for mathematical reasoning.
>
>
> [1] Zenan Li, Zhi Zhou, Yuan Yao, Xian Zhang, Yu-Feng Li, Chun Cao, Fan Yang, Xiaoxing Ma. Neuro-Symbolic Data Generation for Math Reasoning. NeurIPS 2024.
>
> [2] Ning Shang, Yifei Liu, Yi Zhu, Li Lyna Zhang, Weijiang Xu, Xinyu Guan, Buze Zhang, Bingcheng Dong, Xudong Zhou, Bowen Zhang, Ying Xin, Ziming Miao, Scarlett Li, Fan Yang, Mao Yang. rStar2-Agent: Agentic Reasoning Technical Report. Arxiv 2025.
>
> [3] Zenan Li, Zhaoyu Li, Wen Tang, Xian Zhang, Yuan Yao, Xujie Si, Fan Yang, Kaiyu Yang, Xiaoxing Ma. Proving Olympiad Inequalities by Synergizing LLMs and Symbolic Reasoning. ICLR 2025
>
> [4] Weiming Wu, Zi-kang Wang, Jin Ye, Zhi Zhou, Yu-Feng Li, Lan-Zhe Guo. NeSyGeo: A Neuro-Symbolic Framework for Multimodal Geometric Reasoning Data Generation. Arxiv 2025.

---

> > ### Comment · Reviewer_JksY · 2025-11-26
> >
> > Thank you for your thorough response. The authors have clearly addressed my concerns regarding the correctness of the formalization, inference costs, and related work.
> >
> > I have also examined the other reviews and the corresponding author responses, and I did not find any remaining unresolved critical issues.
> >
> > Accordingly, I have raised my score.

---

> > > ### Author Response · Authors · 2025-11-26
> > >
> > > We are deeply grateful for the reviewer's positive assessment of our response and for raising the score. We are pleased that all concerns regarding formalization, costs, and related work are fully resolved. We commit to including additional failure case analyses and a significantly expanded Related Work section in the final paper. Thank you for your constructive feedback.

---

### Official Review · Reviewer_pNN4 · 2025-10-29

**Soundness:** 3
**Presentation:** 3
**Contribution:** 3
**Rating:** 8
**Confidence:** 5

**Summary:**

This paper presents AutoGPS, a framework for automated geometry problem solving that integrates multimodal formalization with deductive reasoning. Unlike traditional text-only or vision-only systems, AutoGPS unifies textual problem statements and geometric diagrams into a consistent symbolic representation, which is then solved through formal logic-based reasoning. The system includes:
1. A multimodal formalization module that parses text and visual diagrams into formal symbolic predicates.
2. A deductive reasoning engine that performs geometric theorem proving through symbolic inference.
3. A data generation and verification pipeline for training and evaluating multimodal geometry reasoning systems.

Experiments demonstrate AutoGPS’s superiority over both LLM-based and neuro-symbolic baselines on geometry reasoning benchmarks, including synthetic and real-world datasets. The approach significantly improves consistency between textual and visual modalities and exhibits better generalization to novel problem types.

**Strengths:**

1. Innovative integration of modalities:
The combination of natural language, diagram understanding, and formal reasoning is both technically challenging and conceptually elegant. The multimodal formalization module is well-motivated and executed with a clear architecture.

2. Clear reasoning pipeline:
The formal-to-symbolic transition is described thoroughly, including explicit steps for entity detection, relation extraction, and logical grounding. The modular design supports interpretability and verifiability.

3. Strong empirical performance:
Experimental results demonstrate substantial improvements over both pure neural models (e.g., GPT-4V, Flamingo) and previous neuro-symbolic systems. The inclusion of accuracy, formal consistency, and visual grounding metrics provides a holistic evaluation.

**Weaknesses:**

1. Limited scalability and automation.
The formalization process still relies partly on rule-based heuristics for entity alignment and relation mapping. It is unclear how the system scales to more complex, noisy, or real-world diagrams with ambiguous geometry.

2. Dataset construction bias.
The training and evaluation datasets appear to be semi-synthetic or curated from well-structured geometry problems. There is limited discussion on how AutoGPS performs on imperfect or non-standard problem statements often found in educational or competition settings.

3. Insufficient discussion of failure cases.
The paper would benefit from analyzing situations where the symbolic representation fails (e.g., misalignment between text and diagram) and discussing how these errors propagate through reasoning.

**Questions:**

As discussed above.

---

> ### Author Response · Authors · 2025-11-25
>
> We thank the reviewer for the insightful comments. We address the concerns as follows:
>
> ## R1. Scalability and Automation
>
> > The formalization process still relies partly on rule-based heuristics for entity alignment and relation mapping. It is unclear how the system scales to more complex, noisy, or real-world diagrams with ambiguous geometry.
>
> We acknowledge that our formalization pipeline includes some lightweight, rule-based heuristics for entity alignment and relation mapping. These heuristics are used primarily for disambiguation and do not form the core of MPF+DSR. To scale to more complex diagrams, extra tools can be integrated into MPF to provide fine-grained information for better performance.
>
> Importantly, the Geometry3K and PGPS9K datasets are derived from real-world high-school mathematics textbooks, providing meaningful real-world reference. Moreover, AutoGPS has also been evaluated on GeoQA and MathVista, demonstrating strong robustness and generalization beyond the original dataset distributions. The experiment results are provided at the end.
>
> ## R2. Dataset Construction Bias
>
> We agree that Geometry3K and PGPS9K represent well-structured problems, but from real-world textbooks. Nonetheless, MPF and DSR are designed to handle diverse problem formulations, and our cross-dataset evaluations on GeoQA and MathVista illustrate that AutoGPS generalizes well to problems outside of the original curated datasets. We will clarify these points in the revised manuscript to highlight the broader applicability of our approach.
>
> ## R3. Failure-case Analysis
>
> We appreciate the suggestion to discuss failure cases. In the current appendix (Appendix 2, Failed Cases Analysis), we present an insightful failure example to inform future research. We acknowledge that this coverage is limited, and in the final version, we plan to include additional meaningful and instructive failure cases to better illustrate potential limitations and guide subsequent work.
>
>
> ## Additional Experiment Results
>
> **MathVista (GPS)**
>
> |              | Method                     | MathVista(GPS) |
> | ------------ | -------------------------- | -------------- |
> |              | Human                      | 48.4           |
> | Close-Source | DreamPRM(o4-mini)          | 95.7           |
> | Close-Source | Step R1-V-Mini             | 89.9           |
> | Close-Source | Kimi-k1.6-preview-20250308 | 91.8           |
> | Close-Source | Doubao-pro-1.5             | 88.9           |
> | Open-Source  | AutoGPS (InternVL3)        | **86.2**       |
> | Open-Source  | Vision-R1-7B               | 82.7           |
> | Open-Source  | Ovis2_34B                  | 84.6           |
>
> **GeoQA**
>
> |                     | GeoQA |
> | ------------------- | ----- |
> | Human               | 92.3  |
> | NGS                 | 57.4  |
> | NGS-Auxiliary       | 60.0  |
> | SCA-GPS             | 64.1  |
> | DualGeoSolver       | 65.2  |
> | FGeo-HyperGNet      | 85.6  |
> | AutoGPS (InternVL3) | 86.2  |

---

### Official Review · Reviewer_U8MK · 2025-10-29

**Soundness:** 3
**Presentation:** 3
**Contribution:** 2
**Rating:** 4
**Confidence:** 4

**Summary:**

This paper addresses the challenges of multimodal understanding and rigorous reasoning in automated geometry problem-solving. It proposes a neuro-symbolic framwork called AutoGPS which is composed by the Multimodal Problem Formalizer (MPF) and the Deductive Symbolic Reasoner (DSR) modules. MPF is used to multimodal understanding, translating the diagram and natural text to rigorous formal languages, while DSR is used to perform deductive reasoning to infer the final answer on a meticulously constructed supergraph. Experimental results demonstrate the effectiveness of the proposed AutoGPS framework.

**Strengths:**

1. The design of the overall neuro-symbolic architecture is reasonable, and the MPF and DSR modules are well defined.
2. The introduction of the multimodal alignment stage in MPF is effective in filling the missing semantic information that is not captured by the pixel-level diagram parser.
3. The experimental results are convincing and demonstrate the effectiveness of the AutoGPS framework.
4. The writing and structure of the paper are clear and easy to follow.

**Weaknesses:**

1. My main concern lies in the originality of this paper. First, in MPF, the text parser $M_t$ should properly cite InterGPS (Line 214). Second, in Line 274, the authors state that "solving algebraic relations remains out of its scope". However, in my understanding, AlphaGeometry adopts a DD + AR symbolic reasoning engine, where AR stands for algebraic reasoning. Therefore, the DSR module appears highly similar to the DD + AR reasoning engine of AlphaGeometry, as well as to the hypergraph expansion component. It would be helpful if the authors could clarify the unique innovations of AutoGPS.
2. Since MPF heavily relies on the diagram parser (PGDPNet), which is trained solely on PGDP and Geometry3K-style data, it may struggle to seamlessly generalize to other styles of geometric diagrams. This limitation further constrains the scalability of AutoGPS.
3. The compared methods are limited. The authors should include comparisons with more recent neuro-symbolic approaches, such as FGeo-HyperGNet [1].

[1] FGeo-HyperGNet: Geometric Problem Solving Integrating FormalGeo Symbolic System and Hypergraph Neural Network

**Questions:**

Besides the weakness above, I wonder what the average time cost of AutoGPS is for solving a geometry problem compared to other baselines.

---

> ### Author Response · Authors · 2025-11-25
>
> ## R1. Originality and Clarification of Writing.
>
> > My main concern lies in the originality of this paper. First, in MPF, the text parser  should properly cite InterGPS (Line 214). Second, in Line 274, the authors state that "solving algebraic relations remains out of its scope". ...
>
> We thank the reviewer for the valuable suggestions and for pointing out ambiguities caused by our writing style.
> We indeed cited *InterGPS* in the same paragraph (Line 221, Lu et al., 2021) and state its limitation and our contribution; however, we agree that the citation was not sufficiently explicit and have revised the manuscript accordingly.
>
> Regarding the relationship to AlphaGeometry, we apologize for the confusion caused by our naming choice (Deductive Reasoning + Algebraic Reasoning), which may appear similar to AlphaGeometry’s DD (Deductive Database) + AR (Algebraic Reasoning) modules. Our intention was to facilitate comparison, but we agree that this unintentionally obscured the conceptual differences. We clarify the distinctions below.
>
> AlphaGeometry' AR module represents **geometric equalities** (e.g., collinearity, concyclicity) as **undirected hyperedges** encoding sets of *equivalent* nodes. Its proof traceback is formulated as a minimum-spanning-tree (MST) problem for undirected hypergraph, which is **NP-hard**, requiring a greedy approximation that **does not guarantee minimal or globally optimal proofs**[1]. To this end, it does **not perform general algebraic computation but geometric equivalence transitions**. **Thus, solving algebraic relations like $\sin(x)=\frac{\pi}{3}$ remains out of its scope.*
>
> In contrast, AutoGPS introduces a **unified deductive–algebraic directed hypergraph** whose edges encode explicit inference dependencies. For Deductive Reasoning, the edge connects *premises and conclusions*. For Algebraic Reasoning, the edge connects *algebraic equation system and its solution*. This formulation **avoids the NP-hard MST traceback entirely**, enabling an **exact algorithm with time complexity $O(n\log n + \Phi)$**[2] for inference and proof extraction. The resulting proof path is deterministic, minimal, and fully traceable.
>
> We will add these clarifications to the DSR module section in the revised manuscript.
>
> [1] Solving olympiad geometry without human demonstrations
>
> [2] Dynamic Shortest Path Algorithms for Hypergraphs
>
> ## R2. Contribution of MPF and Its Scalability
>
> MPF is an **MLLM-based multimodal agent**, and diagram parser is **only one component, constituting less than 1/3 of it.**
> The diagram parser helps the agent to capture the fine-grained geometric information, especially low-level primitives, which complements the limitations of existing MLLMs.
> Moreover, the PGDP-Net can be replaced by other object-detection models. **Replacing the diagram parser does not affect our methodology**. To demonstrate this, we also fine-tuned a YOLO model on PGDP5K to replace PGDP:
>
> |               | Geometry3K | PGDP9K |
> | ------------- | ---------- | ------ |
> | AutoGPS(PGDP) | 81.6       | 81.5   |
> | AutoGPS(YOLO) | 76.3       | 75.9   |
>
> Since PGDP exhibits near-perfect performance on these two datasets, we keep it as the final component. AutoGPS integrated with YOLO also achieves competitive performance, showing that **AutoGPS has strong generalization and scalability and is not limited by PGDP.**
>
> *We will also release our YOLO training code and weights for the community.*
>
> Additionally, our MPF agent introduces multimodal alignment and a feedback–refinement loop that enables the parsing of **higher-level primitives** (e.g., shaded areas, composite regions) and **improves robustness** to incorrect parsing results.

---

> ### Author Response · Authors · 2025-11-25
>
> ## R3. Additional Baselines
>
> We thank the reviewer for pointing out FGeo-HyperGNet. We extract the results from Table 2 of the original FGeo-HyperGNet paper and report the summary performance below:
>
> | Method               | Geometry3K |
> | -------------------- | ---------- |
> | GeoDRL(GT)           | 89.4%      |
> | E-GPS(GT)            | 90.40      |
> | FGeo-HyperGNet       | 91.99%     |
> | AutoGPS(FGeo-Parser) | 92.45%     |
> | AutoGPS(GT)          | 94.5%      |
>
> AutoGPS(FGeo-Parser) is obtained by integrating FGeo-Parser[1] in place of PGDP. The results of FGeo-HyperGNet will be properly cited and added in the revised version.
>
> [1] FGeo-Parser: Autoformalization and Solution of Plane Geometric Problems
>
> ## R4. Timing Results
>
> To address the reviewer’s question regarding runtime efficiency, we report the average solving time per problem using the same hardware configuration as in our main experiments. Since many previous works do not provide timing results, we compare only with publicly available implementations.
>
> | Category        | Method                 | Time (s/problem) |
> | --------------- | ---------------------- | ---------------- |
> | Symbolic        | InterGPS               | 34.7             |
> | Neural–Symbolic | **AutoGPS (InterVL3)** | **53.90**        |
> | MLLM            | Qwen2.5-VL-32B            | 122.9            |
> | MLLM            | Vision-R1-7B           | 242.8            |
>
> We exclude GPT-4o and InterVL3-78B due to API-based inference latency, which is not directly comparable to local execution.
> These results show that **AutoGPS achieves strong performance while maintaining a competitive runtime** among existing systems.

---

### Official Review · Reviewer_bJ89 · 2025-11-01

**Soundness:** 2
**Presentation:** 3
**Contribution:** 2
**Rating:** 4
**Confidence:** 5

**Summary:**

This paper propose AutoGPS, a neural symbolic plang geometry problem solver. AutoGPS consists of a MPF to parse the plane geometry diagram into symbolic language, and further use a DSR to conduct deductive symbolic reasoning based on a hyper graph to solve the problem, and finally obtain the answer by extracting the solution path from the graph. Experiments on Geometry3K and PGPS9K show the proposed method is effective on solving PGPs.

**Strengths:**

1.	The design of the framework is reasonable. AutoGPS first parse the diagram into symbolic languages, further construct a hyper graph of the given problem and search the solution path on the graph to finally get the solution steps, ensure the interpretability.
2.	Comparing to the mentioned previous works in this paper, AutoGPS improved the problem solving accuracy than previous baselines, which show the effectiveness of the proposed method.
3.	In the human evaluation, AutoGPS output stepwise solutions with ideal accuracy, outperformend other pure MLLMs.

**Weaknesses:**

1. As I know, using hyper graph in solving plane geometry probelms (PGPs) was already explored by FGeo-HyperGNet [1], this paper not give appropriate discussion on this previous work, what is the difference of the hyper graph between AutoGPS and HyperGNet. Meanwhile, HyperGNet achieved 91.99% on Geometry3K which is higher than AutoGPS with 81.6.

2. To this end, the contribution of this paper is limited, as the diagram parsing mainly relies on PGDP and the hypergraph has already been used in other work. Despite the author proposing a deductive symbolic reasoning framework to assemble these modules and give stepwise solutions that are easy for humans to read, I think this paper does not satisfy the bar of the ICLR community.

3. The description of symbolic solvers is not appropriate. Indeed, symbolic solvers do not directly output the results without solution steps, like Inter-GPS, which was proposed along with the Geometry3K dataset. Inter-GPS uses a theorem predictor (also has a search algorithm) to predict the theorems at each step, and based on these theorems, to conduct symbolic reasoning on the existing problem conditions to get the final target answer. And the reliable interpretability is one of the advantages of symbolic solvers. The author said symbolic solvers are hard for humans to read, it is easy to tackle by using an LLM to translate the symbolic solution steps into human language, which will be used to understand.

4. Limited experiment dataset, benchmark. This paper only conducted experiments on the Geometry3K and PGPS9K, while actually PGPS9K was expanded from Geometry3K. It is necessary to conduct experiments on popular benchmarks in the PGP solving area, such as Math-Vista Geo, Math-Verse. The current experiments are not convincing.

5. Limited generalization ability of AutoGPS. As mentioned in the above weakness, I am wondering if AutoGPS is hard to solve other domains of PGPs, such as GeoQA, which is also a foundation dataset for the research of solving PGPs, as the diagram parsing module in the AutoGPS is leveraged from PGDP, which is a delicate model designed for parsing diagrams in the style of Geometry3K. If AutoGPS is not able to solve problems in GeoQA and other benchmarks, the scope of this work is not sufficient for ICLR.

Ref:

[1] FGeo-HyperGNet: Geometric Problem Solving Integrating FormalGeo Symbolic System and Hypergraph Neural Network, in IJCAI 2025.

**Questions:**

I would like to see what is the performance of the diagram parsing in AutoGPS by using the feedback and refinement process, comparing to the original PGDP work, which already achieved 99% accuracy on the PGDP5K dataset.

---

> ### Author Response · Authors · 2025-11-25
>
> We thank the reviewer bJ89 for the detailed and insightful comments. Below we address each concern and further clarify the contribution and generalization of AutoGPS.
>
> ## R1. Comparison to FGeo-HyperGNet
>
> >As I know, using hyper graph in solving plane geometry probelms (PGPs) was already explored by FGeo-HyperGNet [1], this paper not give appropriate discussion on this previous work, what is the difference of the hyper graph between AutoGPS and HyperGNet.
>
> We appreciate the reviewer for pointing out FGeo-HyperGNet. After careful examination, we highlight the fundamental differences about hypergraph:
>
> FGeo-HyperGNet uses the hypergraph as a feature input for its neural network (theorem predictor) to **predict the most relevant next theorem** to apply to the current state until the solution is derived.
>
> AutoGPS uses the hypergraph as a **traceability formulation** to ensure every step of the symbolic deduction is **mathematically sound and verifiable**. It further enables a **reliable, minimal and human-interpretable proof** by identifying the most essential steps in the solution.
>
> We will clarify these distinctions in the revised manuscript.
>
>
> > Meanwhile, HyperGNet achieved 91.99% on Geometry3K which is higher than AutoGPS with 81.6.
>
> FGeo-HyperGNet relies on the more powerful FGeo-Parser[1] (BLIP-based diagram parser + T5-based text parser). It can also be seamlessly integrated into our framework. We report our experiment results here.
>
> | Method               | Geometry3K |
> | -------------------- | ---------- |
> | GeoDRL(GT)           | 89.4%      |
> | E-GPS(GT)            | 90.40      |
> | FGeo-HyperGNet       | 91.99%     |
> | AutoGPS(FGeo-Parser) | 92.45%     |
> | AutoGPS(GT)          | 94.5%      |
>
> As shown above, AutoGPS integrated with FGeo-Parser achieves comparable performance (92.45% vs. 91.99%) to FGeo-HyperGNet. On the other hand, with ground-truth formalization, our method can achieve **94.5%** accuracy, outperforming all existing methods.
>
> We will add this extra experimental result in the Appendix, Table 2.
>
> [1] FGeo-Parser: Autoformalization and Solution of Plane Geometric Problems
>
> ## R2. Contribution of MPF and why AutoGPS is not limited by PGDP
>
> MPF is an **MLLM-based multimodal agent**. PGDP is **only one component, constituting less than 1/3 of it and can be seamlessly replaced by other diagram parsers.**
>
> PGDP-Net detects **only low-level primitives** (point/line/circle) on clean diagrams and is replaceable by other models, such as previous diagram parser used in InterGPS, FGeo-Parser or any other fine-tuned object detection model. To demonstrate this, we also fine-tuned a YOLO model on PGDP5K to replace PGDP:
>
> |                   | Geometry3K | PGDP9K |
> | ----------------- | ---------- | ------ |
> | **AutoGPS(PGDP)** | 81.6       | 81.5   |
> | **AutoGPS(YOLO)** | 76.3       | 75.9   |
> | GeoDRL            | 68.4       | -      |
> | E-GPS             | 67.9       | -      |
> | InterGPS          | 63.5       | 66.2   |
>
> Since PGDP exhibits near-perfect performance on these two datasets, we keep it as the final component. *We will also release our YOLO training code and weights for community.*
>
> Additionally, our MPF agent introduces multimodal alignment and a feedback–refinement loop that enables parsing of **higher-level primitives** (e.g., shaded areas, composite regions) and robustness to incorrect parsing results.
>
> Thus, AutoGPS provides a broader and more reliable formalization module than approaches based solely on PGDP.
> ## R3. Output granularity and interpretability of symbolic solvers
>
> (1) InterGPS does produce explicit reasoning steps, but only as theorem ID sequences **without the associated premises or conclusions**, making it hard to interpret or faithfully translate, even for human experts. Interpreting with LLM will **introduce noise and reduce the reliability**.
>
> (2) InterGPS **uses a global equation system**, where all derived equations from theorem applications are aggregated, and **the final solution always relies on a non-minimal subset**, which significantly reduces interpretability.
>
> (3) AutoGPS instead extracts a **minimal, redundancy-free hyper-path**, where each step contains explicit premise–conclusion pairs, producing concise and human-readable proof sequences.

---

> ### Author Response · Authors · 2025-11-25
>
> ## R4. Additional experiments
>
> >W4. Limited experiment dataset, benchmark... It is necessary to conduct experiments on popular benchmarks in the PGP solving area, such as Math-Vista...
>
> >W5. Limited generalization ability of AutoGPS... If AutoGPS is not able to solve problems in GeoQA and other benchmarks, the scope of this work is not sufficient for ICLR.
>
> Following the reviewer’s suggestion, we evaluate AutoGPS on two widely used PGP benchmarks.
>
> **MathVista (GPS)**
>
> AutoGPS (InternVL3) achieves **86.2%**, ranking **1st among existing open-source methods** and competitive with several close-source models.
>
> |              | Method                     | MathVista(GPS) |
> | ------------ | -------------------------- | -------------- |
> |              | Human                      | 48.4           |
> | Close-Source | DreamPRM(o4-mini)          | 95.7           |
> | Close-Source | Step R1-V-Mini             | 89.9           |
> | Close-Source | Kimi-k1.6-preview-20250308 | 91.8           |
> | Close-Source | Doubao-pro-1.5             | 88.9           |
> | Open-Source  | AutoGPS (InternVL3)        | **86.2**       |
> | Open-Source  | Vision-R1-7B               | 82.7           |
> | Open-Source  | Ovis2_34B                  | 84.6           |
>
> **GeoQA**
>
> AutoGPS achieves **86.2%**, outperforming NGS, SCA-GPS, and DualGeoSolver, and is comparable to FGeo-HyperGNet (85.6%)
>
> |                     | GeoQA |
> | ------------------- | ----- |
> | Human               | 92.3  |
> | NGS                 | 57.4  |
> | NGS-Auxiliary       | 60.0  |
> | SCA-GPS             | 64.1  |
> | DualGeoSolver       | 65.2  |
> | FGeo-HyperGNet      | 85.6  |
> | AutoGPS (InternVL3) | 86.2  |
>
> These results demonstrate AutoGPS’s strong cross-dataset generalization ability, which stems from the MLLM-based MPF agent for more general and robust problem formalization.
>
>
> ## R5. Effectiveness of the feedback-and-refinement mechanism
>
> >I would like to see what is the performance of the diagram parsing in AutoGPS by using the feedback and refinement process, comparing to the original PGDP work, which already achieved 99% accuracy on the PGDP5K dataset.
>
> More precisely, PGDP-Net achieves 99\% **Likely Same** and 84.7\% **Totally Same** accuracy on the PGDP5K dataset. **These small but non-negligible inconsistencies often propagate and cause symbolic solvers to fail.** The refinement mechanism specifically **corrects such subtle errors**.
>
> We conducted experiments to quantify the effect of refinement. Specifically, we compare our Refine@5 mechanism (a single solution attempt with *a maximum of five feedback and refinement iterations*) with a naive Pass@5 strategy (*five independent forward passes* without refinement or feedback, *seen as correct if any pass is correct*). The evaluation is based on *strict Completion Accuracy*.
>
> | Method       | Geo3K     | PGPS9K    | GeoQA     |
> | ------------ | --------- | --------- | --------- |
> | **Refine@5** | **71.2%** | **73.3%** | **71.9%** |
> | Pass@5       | 68.2%     | 70.0%     | 67.3%     |
>
> Refinement yields **+3.0%, +3.3%, +4.6%** improvements while requiring **only 0.36–0.51 additional forward passes on average**, significantly fewer than those needed for Pass@5 (5 passes). This confirms that refinement is both **effective** and **compute-efficient**.

---

### Meta-Review · Area_Chair_rssy · 2025-12-25

**Summary:**

The common concerns of the reviewers are about the tests on more datasets, the comparison with the SOTA methods (FGeo-HyperGNet in particular) and the inference cost. The authors have given the pointwise feedbacks with more tests on Mathvista and GeoQA, quantitive comparison with FGeo-HyperGNet, and inference overhead. The rebuttal is solid and I decide to accept the paper.

**Reviewer Concerns:**

Most concerns, I believe, are well addressed.

**Reviewer Scores:**

Average score is 5.5 and Reviewer JksY is expected to raise the score. Most concerns are addressed using the concrete experimental results and the reviewers are supposed to maintain or increase the rating.

---

### Decision · Program_Chairs · 2026-01-26

Accept (Poster)